



**Impact assessment of terrestrial and marine air-mass on the**
**constituents and intermixing of bioaerosols over coastal atmosphere**
Qun He[a], Zhaowen Wang[a], Houfeng Liu[a], Pengju Xu[a], Rongbao Duan[a], Caihong Xu[b],
Jianmin Chen[b], Min Wei[a,b,*]
[a] College of Geography and Environment, Shandong Normal University, Ji'nan 250014, China
[b] Shanghai Key Laboratory of Atmospheric Particle Pollution and Prevention (LAP3), Fudan
Tyndall Centre, Department of Environmental Science & Engineering, Fudan University,
Shanghai 200433, China
* Correspondence to: Min Wei (minwei@sdnu.edu.cn)
**Abstract**:
Coastal environments provide an ideal setting for investigating the intermixing
processes between terrestrial and marine aerosols. Fine particulate matter ($PM_{2.5}$)
samples collected from a coastal location in Northern China were categorized into
terrestrial, marine and mixed air masses. Chemical and biological constituents during
the winter heating season in 2018, including the water-soluble ions (WSIIs), metallic
elements, and bacterial and fungal aerosols, were investigated. Terrestrial air masses
constituted a larger proportion of 59.94%, particularly during severe air pollution
episodes (up to 90%), exhibiting higher concentrations of $PM_{2.5}$ (240 μg/m³) and
carrying more water-soluble ions and metal elements. A relative shift towards marine
air-mass with respect to pollution elimination stage was observed. The terrestrial air
mass harbors more animal parasites or symbionts, and human pathogens from
anthropogenic emission, such as *Deinococcus*, *Sphingomonas*, *Lactobacillus*,
*Cladosporium* and *Malassezia*. In comparison, saprophytic bacteria and fungi, such as



hydrocarbon degradation and gut bacteria from *Comamonas*, *Streptococcus*,
*Novosphingobium*, and *Aerococcus*, saprophytic *Aspergillus*, were the most prevalent
species in marine air mass. Mixed air-mass revealed the intermixing processes of
terrestrial and marine sources. This is a consequence of the amalgamation of
microorganisms from both terrestrial soils, animals, plants, and marine environments
during transportation. Correlation analysis suggested a higher correlation between
microorganisms and continental air mass, such as $K^+$, $Mg^{2+}$, and $Ca^{2+}$ from soil dust.
Present study on constituents and amalgamation of bioaerosols over coastal atmosphere
encompassing distinct airmasses presume critical importance in comprehending the
terrestrial and marine air mass transport, intermixing processes and health implications.
**Keywords:** $PM_{2.5}$, water-soluble ions, terrestrial air mass, marine air mass, bioaerosols



## 1. Introduction

Bioaerosols, encompassing bacteria, fungi, viruses, pollen, cellular debris, etc., are crucial aerosol particles in the atmosphere. These particles, ranging in size from 0.1 to 100 μm, are abundant in microorganisms or biomolecules that demonstrate life activity (Górny, 2020; Urbano et al., 2011; Vishwakarma et al., 2023). Notably, bacterial aerosols which can be either free-floating or attached to airborne particles, typically measure between 0.3 and 10 μm (Tamer Vestlund et al., 2014). Such characteristics allow for extended residence times and a heightened likelihood of long-distance transport from local to distant regions within the atmospheric circulation (Galbán et al., 2021; Smets et al., 2016). Fungal aerosols are prevalent in Earth's near-surface atmosphere, with their origins being diversely found in water, soil, and plants (Eduard, 2008). Furthermore, the presence of pathogenic bacteria and fungi in the atmosphere may lead to significant penetration efficiency within the human respiratory system (Fakunle et al., 2021; Jiang et al., 2022; Smets et al., 2016).

Relevant studies have demonstrated that the concentration and community structure of bacterial and fungal aerosols are significantly influenced by air emission sources, atmospheric circulation, meteorological conditions, geographical and topographical factors (Li et al., 2019; Liu et al., 2021; Núñez et al., 2019; Núñez et al., 2021). The geographical and topographical factors, such as terrestrial and marine environments exhibit significant differences in bioaerosol sources and pollution characteristics. Studies focusing on continental environments, particularly in large inland cities like Beijing (Zhang et al., 2019; Zhang et al., 2022), Xi'an (Wang et al., 2020; Yao et al., 2019), Jinan (Wei et al., 2020) , and Nanjing (Hu et al., 2020) during severe air pollution, have revealed that continental pollutant emissions significantly contribute to air pollution. Furthermore, transboundary transportation of terrestrial air masses on hazy



days plays a crucial role in the progression of regional heavy pollution (Xie et al., 2020).
The ocean serves as both a potential source and sink for airborne microorganisms
(Archer et al., 2020; Mayol et al., 2017), exhibiting the intricate interplay between
continental communities and their transmission across oceanic regions (Bhangar et al.,
2015; Cho and Hwang, 2011; DeLeon-Rodriguez et al., 2013). Bioaerosols from the
oceans maybe influenced by long-distance transport from continental sources, such
as plants and human pathogens (Elbert et al., 2007; Sharoni et al., 2015). Studies have
shown that the concentration and diversity of bacterial and fungal aerosols from marine
are typically lower than those derived from continental sources (Cao et al., 2024; Shi et
al., 2022; Xue et al., 2022). Xu et al. (2019) undertook a thorough investigation of
bacterial abundance in Mt. Tai, China. Their findings indicate that variations in airmass
from diverse sources could potentially influence the chemical composition of $PM_{2.5}$.
This in turn prompts shifts in bacterial groups. Limited
studies have examined the impacts of terrestrial and marine air masses on chemical
constituents and microbial aerosols (Aswini and Hegde, 2021; Lang-Yona et al., 2022;
Shi et al., 2022). Generally, the chemical aerosols can be affected by various sources of
air masses, which may include local aerosols or remotely transported aerosols. However,
it remains unclear whether different air trajectories contribute to the formation of
bacterial communities within these particles. There is a dearth of research focused on
determining the chemical and biological composition of coastal cities affected by these
air masses.
Coastal atmospheres frequently display a complex amalgamation of terrestrial and
marine aerosols, with their characteristics being markedly influenced by the origins of
the air masses. Therefore, the costal aerosols provide the ideal conditions for
understanding the mixing processes taking place between natural and anthropogenic air



masses from terrestrial and marine, respectively. Meteorological factors as well as
changes in pollutant concentrations and compositions exhibit significant variability
under different air masses. The average abundance of bacterial and fungal spores in the
atmospheric boundary layer over land is approximately $1.9\times10^4$ cells/m$^3$ and $2.4\times10^4$
cells/m$^3$, respectively (Mayol et al., 2014; Spracklen and Heald, 2014). Marine
microbial aerosols can be released from ocean micro-surface or transported from land
(Prospero et al., 2005) and settle thousands of kilometers away from their source of
release (Mayol et al., 2014), exerting significant impacts on ecological and climate
systems. Atmospheric aerosols traversing marine regions have been documented to
contain marine bacteria from the *Cyanobacterial* and *α-Proteobacterial* classes, which
are predominantly observed in air samples (Aller et al., 2005). Gong et al (2020a)
conducted an examination of the microbial composition along Qingdao's coastlines,
revealing a higher proportion of bacteria to total microorganisms in samples from
continental sources compared to marine sources. Xu et al (2020b) studied the diversity
of bacterial populations in PM$_{2.5}$ across urban and rural areas of Shanghai, finding that
airborne microbial communities over coastal cities are more influenced by long-
distance transport than those inland. Some marine bacteria persist in aerosols after land
transportation (Raman and Wagner, 2011). Air samples taken at high altitude zones in
coastal Europe and Japan demonstrated the continental transportation of marine
microorganisms (Maki et al., 2014). The introduction of marine bacteria into
tropospheric free space may modify the airborne microbial composition in continental
regions (Amato et al., 2007; Cáliz et al., 2018; Cho and Hwang, 2011; Maki et al., 2014;
Polymenakou et al., 2008).
Weihai, a coastal city situated at the confluence of the eastern Asia continent and the
Northwest Pacific Ocean, is prone to the impact of marine and terrestrial air masses.



Typically, Weihai maintains low pollutant emissions with an annual average $PM_{2.5}$
concentrations below 35 μg /m$^3$ throughout the year. However, during winter and spring
periods, regional air pollution intensifies, leading to severe air quality issues. This is
due to increased inter-regional transportation, resulting in daily average $PM_{2.5}$
concentrations greater than 150 μg/m$^3$ (Wei et al., 2020). In this study, we conducted an
integrated atmospheric observation experiment to examine the potential impact of
terrestrial and marine air-mass on the constituents and amalgamation of bioaerosols
over the coastal atmosphere. This approach allows for a comprehensive exploration of
the effects of sea-land air mass exchange on the spatial and temporal distribution of
aerosols, as well as potential intermixing processes in coastal regions.
**2.   Materials and methods**
**2.1 Sample collection**

The sampling site was situated at the national air sampling station of Shandong

University (37.53°N, 122.06°E), approximately 1-2 km from the coast (Fig. S1). The
sampling platform was mounted on the rooftop of a building, positioned about 15 m
above ground level, and devoid of any significant obstructions. $PM_{2.5}$ samples were
gathered between January to March, 2018, during the winter heating season in
northern China. Two parallel $PM_{2.5}$ sampler with a particle size of 2.5 ± 0.2 μm were
utilized equipped with 47 mm Quartz membrane for the collection and analysis
of atmospheric $PM_{2.5}$, inorganic ion, metal elements, and microbes. Prior to use, the
Quartz membranes were cauterized in a muffle furnace at 450°C for 6 hours to eliminate
carbonaceous and contaminant materials. Samples were collected twice a day
(07:00~19:00; 19:00~07:00 the following day). Ultimately, 300 $PM_{2.5}$ samples were
obtained, out of which 40 samples were chosen for subsequent analysis based on



air mass categories. Blank samples were obtained using membranes without sampling,
and all these samples were stored at -80°C until further analysis.
Meteorological parameters, such as air temperature, relative humidity, wind direction,
and wind speed, were monitored in situ utilizing a PC-4 automatic weather station (PC-
4, JZYG, China). The hourly concentrations of $PM_{2.5}$, $PM_{10}$, CO, $SO_2$, $NO_2$, and $O_3$
were systematically retrieved from the National Ambient Air Quality Monitoring
System (http://www.cnemc.cn/).
The mass concentration of $PM_{2.5}$, water-soluble ions, and metal elements were
quantified post sampling. The membranes were meticulously weighed utilizing a
Mettler XP-6 balance with an accuracy of $10^{-6}$ g. Prior to the weighing, the membranes
were maintained in a controlled environment with consistent temperature and humidity
for a duration of 24 hours. Ion chromatography (ICS-2100, Chameleon 6. 8, AS-DV
autosampler Thermo Fisher) was employed to ascertain the presence of water-soluble
ions such as $Na^+$, $K^+$, $Ca^{2+}$, $Mg^{2+}$, $Cl^-$, $NO_3^-$, $SO_4^{2-}$, and $NH_4^+$. These ions were extracted
via ionized water ultrasonication, subsequently separated through anion or cation
column exchange, and identified using a conductivity detector. Metallic elementals
were scrutinized using microwave digestion extraction (ETHOS ONE, Milestone), with
the concentrations determined by ICP-MS or ICP-OES (Thermo Fisher). This analysis
encompassed elements as Al, Fe, Ti, Mn, Co, Ni, Cu, Zn, Ga, Sr, Cd, Sn, Sb, Pb, V, Cr,
and As.
**2.2 Air mass clustering and classification**
The potential sources and transport pathways of air mass were examined using the
MeteoInfo      backward      trajectory      model      (MeteoInfo      3.7.4      –      Java,
http://www.meteothink.org/downloads/index.html) developed by the Chinese Academy



of Meteorological Sciences. For each sample, backward trajectories were simulated one-
hour intervals and estimated over a 24-hour period. The meteorological data were
sourced     from     GDAS1     (ftp://arlftp.arlhq.noaa.gov/pub/archives/gdas1/).
Backward trajectories of air masses at an altitude of 500 meters were categorized and
clustered, with daily plots illustrating these trajectories.  In this study, we defined a
marine air-mass sample if more than 90% of the masses originated from the ocean;
terrestrial air-mass sample if more than 90% originated from the continent; and mixed
if the proportions of terrestrial and marine air masses were similar or accounted for
more than 40% of the total air masses.
**2.3 DNA extraction and qPCR amplification**
Microbial genomic DNA in $PM_{2.5}$ samples were procured from filters utilizing the
Fast-DNA$^{TM}$ SPIN kit for soil (MoBio Laboratories, Carlsbad, CA, USA). The
extracted DNA was measured via a Nanodrop spectrometer (Nanodrop 2000, Thermo
Scientific USA) to ascertain the concentration. The quantitative polymerase chain
reaction (qPCR) was employed to identify bacterial 16S rRNA and fungal ITS gene
copy numbers in $PM_{2.5}$, as well as to estimate the count of bacteria and fungi per cubic
meter of air. The bacterial 16S V3-V4 variable region was selected for PCR
amplification using primer 338F (5'-ATCTACGGGGGGCAGCAG-3') and 806R
(5'GGACTACHVGGGTWTCTAAT-3') (Masoud et al., 2011). The fungal ITS region
were amplified using the primers ITS1 (5′-CTTGGTCATTTAGAGGAAGTAA-3′) and
ITS2 (5′-GCTGCGTTCTTCATCGATGC-3′) (Liu et al., 2021).
The PCR amplification conditions comprised an initial denaturation at 95°C for
duration of 5 minutes, succeeded by 30 seconds at 95°C, 30 seconds at 50°C, and then
35 cycles at 72°C for 40 seconds, followed by a final extension phase at 72°C



for seven minutes to ensure comprehensive amplification. Fluorescent signals were
gathered during this extension phase. For each sample, qPCR was conducted in
triplicate, with ultrapure water serving as a negative control to identify potential PCR
contamination. Standard curves were constructed using *E. coli* harboring the 16S rRNA
gene and *Streptomyces* plasmids containing the ITS gene. Additionally, gradient
dilutions of these plasmids were performed, ranging from $10^2$-$10^7$ copies/µL. The FTC-
3000 real-time quantitative PCR system was employed for standard curve construction
and data processing.
**2.4 16S rRNA and ITS gene sequencing and data processing**
Similarly, the V3-V4 region of the bacterial 16S rRNA and the fungal ITS1 gene
were targeted for PCR amplification utilizing barcode-specific primers 338F-806R and
ITS1F-ITS2, respectively. To ensure optimal amplification efficiency and precision,
a high-fidelity enzyme (Phusion® High Fidelity PCR Master Mix from NewEngland
Biolabs) in conjunction with GC buffer was employed during PCR amplification.
This procedure entailed a pre-denaturation step at 98°C for one minute, followed by 30
cycles of 98°C for ten seconds; 58°C for bacteria (56°C for fungi) for 30
seconds; and 72°C for an additional 30 seconds. This cycle was repeated seven times,
concluding with a final extension at 72°C for five minutes, after which the samples
were stored at 4°C. Following amplification, the samples were purified using the
Agencourt Ampure XP kit (from Beckman Coulter, Brea, CA, USA). Subsequently,
they were combined to achieve equimolar concentrations and analyzed on the Illumina
MiSeq platform provided by Illumina, Inc. in San Diego, CA.
Following sequencing, the barcode sequences of each sample were extracted and
subsequently stored in fastq format utilizing the QIIME toolkit (Caporaso et al.,

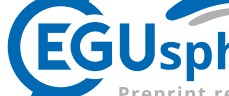



2010). Sequences shorter than 200 base pairs, with mass of less than 25, and containing
ambiguous bases underwent quality control screening using Trimmomatic (Bolger et
al., 2014) and Mothur (Schloss et al., 2009). The assembled sequences were then de-
duplicated and trimmed to equal length. De-chimerized sequences were eliminated
(Edgar, 2013), and sequences were clustered into operational taxonomic units (OTUs)
using UPARSE at a 97% identity threshold. Subsequently, individual OTUs were
removed. Taxonomic assignments were determined using a Basic Local Comparison
Search Tool (BLAST) search against the bacterial SILVA Archived SSU Ref NR 99 132
dataset and fungal UNITE ITS reference v7.2 dataset. The original raw sequences were
deposited in the Sequence Read Archive under the accession number PRJNA1096829.
**2.5 Statistical analysis**
The bacterial community functional analysis was conducted using FAPROTAX, a
manually constructed database that maps prokaryotic taxa to metabolic or other
ecologically functions, such as sulfur, nitrogen, hydrogen, and carbon cycling.
FUNGuild (Fungi Functional Guild) facilitates the taxonomic analysis of fungal
communities by employing microecological guilds. Samples from different air-mass
samples were examined for intergroup species variability, based on community
abundance data. This was achieved using rigorous statistical methods to identify species
demonstrating differences in abundance within the microbial communities of different
groups, and hypothesis testing to evaluate the significance of these observed differences.
Statistical analysis, including Analysis of variance (ANOVA) and Kruskal Wallis tests,
were employed to discern bacteria with varying abundances between samples and
groups. A $p$ value less than 0.05 was considered significant. ANOVA was used to
analyze variation in a response variable measured under conditions defined by discrete



factors (Martin G, 2008). The Kruskal-Wallis test determines whether there is a
statistically significant difference between the medians of three or more independent
groups (Kassambara, 2019). The disparities and primary influencing factors on
microbial community between terrestrial and marine air masses were assessed using the
Mantel correlation analysis, a method predominantly utilized in ecology
to examine the relationship between community and environmental variables. The
statistical was determined using the Spearman's rank correlation coefficient, with
significance levels set at $p<0.05$ and $p<0.01$.
**3. Results and Discussion**
**3.1 Air mass backward trajectory**

The MeteoInfo backward trajectory model was employed to simulate the trajectories

of air masses at an altitude of 500 m in Weihai over 24-hour period from January to
March 2018, identifying and classifying potential sources of air mass transport (Fig.
S1). The terrestrial air mass accounted for 59.94%, exhibiting an average $PM_{2.5}$
concentration of $36.15\pm26.52$ μg/m³. Severe air pollution episodes occurred on January
20, March 19, and March 24, with $PM_{2.5}$ concentration reaching 240 μg/m³, 153 μg/m³,
and 119 μg/m³, respectively. During reginal air pollution, the terrestrial air masses
primarily influenced Weihai were typically originating from the Beijing-Tianjin-Hebei
region and the surrounding areas. The significant ownership of motor vehicles is
identified as a major source of nitrogen oxides (Yang et al., 2018). Furthermore, the
primary contributors to emissions in the terrestrial air mass of this region are the dense
population, industrial and agricultural activities (Wei et al., 2019).

The marine air masses mainly originate from the Yellow Sea, traversing the marginal

sea of the western Pacific Ocean before making contact with the land prior to reaching





the study area. A total of 14% of the sampling days were influenced by these marine air
masses, resulting in an average $PM_{2.5}$ concentration of 23.99±11.00 μg/m³. Mixed air
masses, characterized by simultaneous influence from the northwestern winds of Inner
Mongolia and the offshore air masses of the Yellow Sea, accounted for 27% of all
affected sampling days. These mixed air masses yielded an average $PM_{2.5}$
concentration of 45.11±12.69 μg/m³. In comparison, under the influence of mixed air
masses, pollutant concentrations were notably high. During the spring dust season,
there is a notable increase in the proportion of mixed air mass. This elevated
concentration of particulate matter correlates with the transmission of sand dust (Xie et
al., 2020).
Three heavy pollution episodes were examined to investigate air mass shifts during
pollution (Fig. 1, Fig. S2). Generally, the initiation and development stages of pollution
events were predominantly characterized by terrestrial and mixed air masses.
Conversely, during the stage of pollution mitigation, marine air masses were the
primary contributors. For instance, the pollution episode I occurred from January 19 to
January 21. During this period, the western terrestrial air mass was predominantly
responsible for initiating the pollution, evidenced by a $PM_{2.5}$ concentration of 51.35
μg/m³. As the pollution progressed, it transitioned into a mixed air mass with a $PM_{2.5}$
concentration peaking at 240 μg /m³. This change was accompanied by significant
increases in WSIIs and elemental concentrations. The introduction of marine air mass
from the west led to the elimination of the pollution, resulting in an average $PM_{2.5}$
concentration reduction to 7.92 μg /m³. This reduction was also associated with lower
WSIIs and elemental concentrations, due to the scavenging effect of the marine air mass.



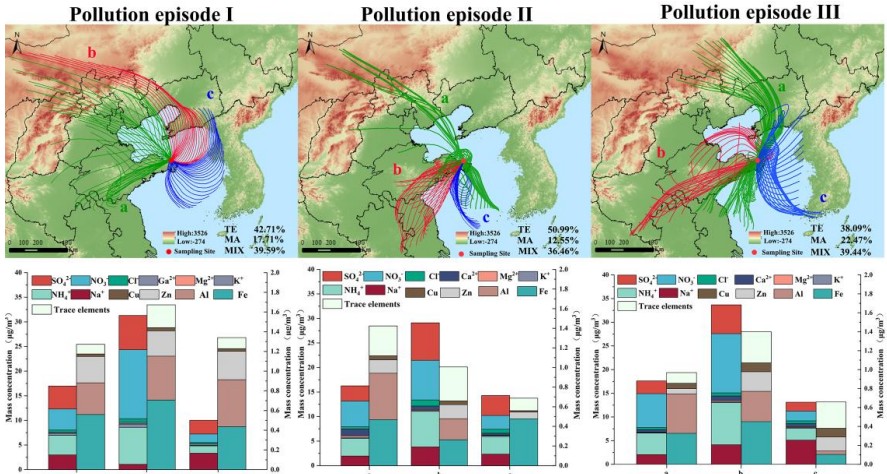

**Fig. 1 Transformation of terrestrial and marine air masses of three severe air pollution episodes. A, the pollution initiation; b, the pollution development; c, the pollution elimination. TE, terrestrial air mass; MA, marine air mass; MIX, mix air mass.**

**3.2 Water-soluble ions and metal elements concentration**

During the sampling period, the daily average concentration of water-soluble ions in $PM_{2.5}$ was 22.92±12.19 μg/m³, exhibiting a range of variation from 4.58 to 78.14 μg/m³. Notably, $NO_3^-$ had the highest concentration (26.94%, ranging from 6.4% to 52.6%), succeeded by $SO_4^{2-}$ (21.94%, between 9.4% and 33.4%) and $NH_4^+$ (20.26%, from 5.8% to 35.6%). The concentrations of $SO_4^{2-}$, $NO_3^-$ and $NH_4^+$, constituted the majority of total ion concentration in other studies conducted in Beijing, Shanghai, and Guangzhou (Hu et al., 2014; Pathak et al., 2008; Zhou et al., 2012). The concentrations of $NO_3^-$, $SO_4^{2-}$, and $NH_4^+$ were significantly influenced by both terrestrial and mixed air masses, with the latter exhibiting a more pronounced effect ($NO_3^-$, 10.65±3.26 μg/m³; $NH_4^+$, 7.39±3.30 μg/m³; $SO_4^{2-}$ 6.76±1.77 μg/m³) (Fig. S3). In marine air-mass samples, a notably lower concentration of water-soluble ions was observed, with the concentration of 13.01±7.43 μg/m³, 27.94±13.61 μg/m³ and 30.38±11.38 μg/m³ in marine, terrestrial and mixed air masses, respectively.





A high concentration of $Na^+$ was observed, with a range that from $3.15 \pm 1.69$ μg/m$^3$,
which accounted for 14.47% of the total water-soluble ion. The concentrations of $Na^+$
and $Mg^{2+}$ did not significantly differ in the three types of air-mass samples. These two
ions are typical components of sea salt. The average ratio of $Mg^{2+}/Na^+$ was found to be
0.11, which closely aligns with the value of 0.12 in seawater. This suggests potential
sources from marine environments (Sun et al., 2022). The concentrations of $K^+$
0.24±0.20 μg/m$^3$ and 0.26±0.10 μg/m$^3$ in the terrestrial and mixed air-mass samples,
and was twice as high as those in the marine air-mass samples (0.11±0.05 μg/m$^3$), which
suggests an important contribution from anthropogenic emissions. The concentrations
of $Cl^-$, and $Ca^{2+}$ in both terrestrial and marine air mass samples exhibited similarity.
However, a marked increase was observed in mixed air masses. $Cl^-$ mainly comes from
sea salt, coal and biomass combustion, and $Ca^{2+}$ is mainly affected by sand dust in
spring (Liang et al., 2021). The coastal city was more affected by sea salt, and the coal-
fired power plants in winter heating season. The high concentration in mixed air mass
were associated with the sea salt, coal combustion, dust events, and construction
activities (Sun et al., 2021).
The concentration of various elements in different air-mass samples are depicted in
Fig. S3. The top ten elements identified in PM$_{2.5}$ were found to be Fe, Al, Zn, Cu, Sn,
Pb, Mn, Ti, Ni, and V, in descending order. The metal elements were categorized into
macro and trace elements. The macro metals, specifically Iron (Fe), Aluminum (Al),
and Zinc (Zn), constituted a significant proportion of the total heavy metal elements,
accounting for 34%, 25%, and 23% respectively. In general, the concentration of both
macro and trace metal elements in marine air masses was found to be lower than that in
terrestrial and mixed air masses. The V/Ni ratio serves as an indicator of ship emission
influence (Celo et al., 2015; Viana et al., 2009). A ratio of V/Ni exceeding 0.7 suggests



a substantial impact from ship emission sources, typically used as an indicator of ship
emission influence in coastal cities, in conjunction with the trajectories of air mass
transport. (Zhang et al., 2014). The V/Ni ratio in marine air-mass samples was found to
be 0.78, a value significantly higher than that of both terrestrial and mixed air masses.
**3.3 Microbial community over coastal atmosphere**
Airborne bacterial and fungal concentration in PM$_{2.5}$ were $1.99\pm1.46\times10^5$ cells/m$^3$
and $3.39\pm1.10\times10^4$ cells/m$^3$, respectively. A high concentration was observed in
terrestrial air-mass samples, with the average value of $4.72\pm3.93\times10^5$ cells/m$^3$ and
$6.37\pm1.70\times10^4$ cells/m$^3$ for bacteria and fungi, respectively. The terrestrial air masses
came from the inland areas, which carried more microorganisms from anthropogenic
activities and natural sources with high resistant to high temperatures, dryness, and
strong ultraviolet rays (Gong et al., 2020). Microbial concentration in marine air-mass
samples was significantly lower, with the average concentrations of bacteria and fungi
being $4.91\pm1.82\times10^4$ cells/m$^3$ and $6.15\pm3.09\times10^3$ cells/m$^3$.

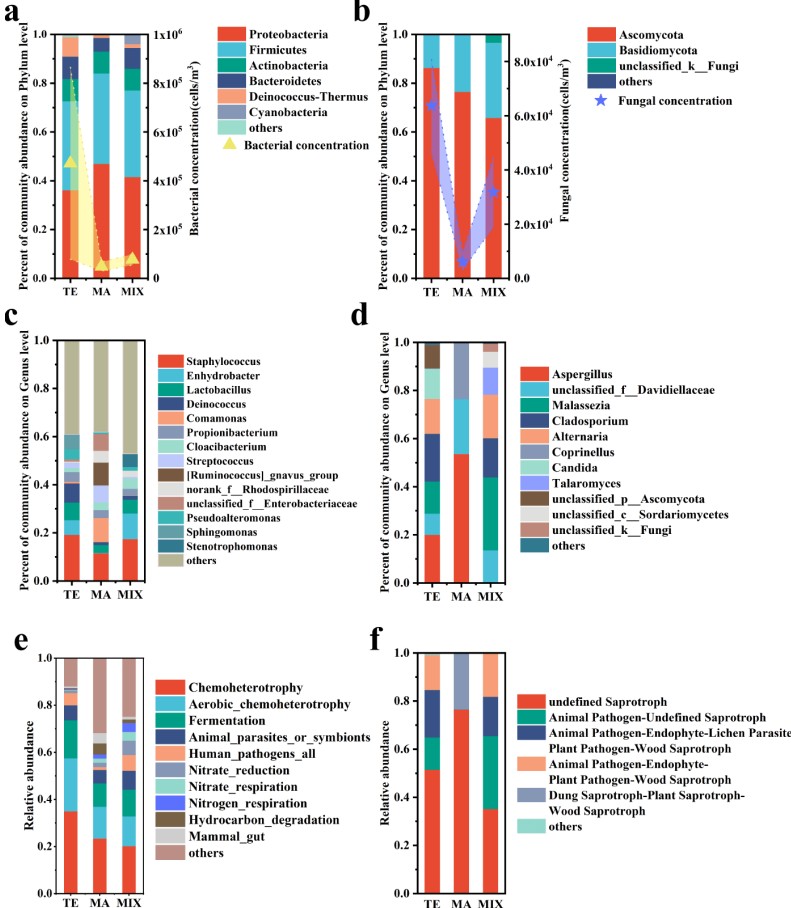

**Fig. 2. Bacterial and fungal species and function analysis influenced by different air masses. Bacterial and fungal concentration, main phylum (a), (b), Bacterial and fungal genus (c), (d), and bacterial and fungal community function (e), (f) are indicated.**

The microbial community structure exhibited significant variation influenced by marine and terrestrial air masses (Fig. 2). Predominantly, *Proteobacteria* (40.06%), *Firmicutes* (36.30%), *Actionbacteria* (8.97%), *Bacteroidota* (8.29%), and *Deinococcus-Thermus* (4.59%) were identified as the most abundant bacteria. Notably, *Actionbacteria* and *Deinococcus-Thermus* were found in high concentrations in samples from terrestrial air masses. In particular, *Deinococcus-Thermus* demonstrated a relative abundance of 7.9% in terrestrial air mass samples, significantly surpassing



that of marine (1.2%) and mixed air mass samples (1.5%). As a radiation-resistant
bacterium, *Deinococcus-Thermus* is capable of withstanding harsh environmental
conditions (Callegan et al., 2008; Rainey et al., 2007). The majority of these bacteria
possess the ability to produce spores or spore germs. These spores are capable of
withstanding harsh conditions such as low temperatures, aridity, and radiation during
long-distance transmission processes, thereby ensuring their survival throughout this
process (Griffin, 2003). *Cyanobacteria* exhibit a higher concentration in both terrestrial
and mixed air-mass samples. As typical soil bacteria found in desert environments, they
are prone to forming soil crusts in arid regions (Li et al., 2014; Li et al., 2013).
*Proteobacteria* was the most prevalent taxon in marine air-mass samples. The
predominant metabolic activity in deep-sea sediments is attributed to this group, with
major taxa being found in marine sediments (Huang et al., 2021).
The predominant bacterial genera included *Staphylococcus* (21.02%),
*Enhydrobacter* (6.43%), *Lactobacillus* (6.03%), *Deinococcus* (4.56%), and
*Propionibacterium* (3.48%). Terrestrial air masses and mixed air masses had similar
bacterial community composition. The abundance of bacteria such as *Staphylococcus*,
*Enhydrobacter*, *Lactobacillus*, *Deinococcus* is higher influenced by terrestrial and
mixed air masses. *Staphylococcus* is a pathogenic bacterium widely found in human
skin, nasal cavity, throat, and intestines (Cheung et al., 2021). Relative abundance of
bacteria such as *Comamonas*, *Streptococcus*, *Ruminococcus*, and *Enterobacteriaceae*
were higher in marine air-mass samples. *Comamonas* is a gram-negative bacillus,
inhabiting in natural soil, freshwater, and animal gut. They have also been isolated from
industrial environments, such as activated sludge and contaminated soils, as well as
from hospital environments and clinical samples. *Comamonas* is associated with
environmental bioremediation, and is considered an important environmental bacterium



rather than a human pathogen (Ryan et al., 2022). *Streptococcus* are mostly found in
the oral and gastrointestinal tracts of a variety of mammals and have not been shown to
play a role in human infections to date (James et al., 2015). *Ruminococcus*, and
*unclassified Enterobacteriaceae* are gut microorganisms that may be related to the
marine fish and other animal gut microbes. Most bacteria carried by marine air masses
with high abundance are saprophytic in nature.

The dominant fungal phyla were *Ascomycota* (77.29%) and *Basidiomycota* (21.58%),

which were similar to the previously studies (Du et al., 2018; Liu et al., 2019; Zeng et
al., 2019). Fungal community influenced by terrestrial and mixed air masses were quite
similar, with relatively higher abundances of opportunistic pathogens such as
*Malassezia*, *Alternaria*, *Cladosporium*. In contrast the saprophytic *Aspergillus*,
*Davidiellaceae* and *Coprinellus* were abundant in marine air-mass samples.
**3.4 Community disparities influenced by terrestrial and marine air masses**

The community disparities influenced by terrestrial, marine, and mixed air masses

was conducted in Table S2, Table S3, Fig. 3 and Fig. 4. The bacterial enrichments
observed in the samples from terrestrial and mixed air-masses included *Deinococcus*,
*Lactobacillus*, and *Sphingomonas*. *Deinococcus* is known to tolerate high radiation and
adapted to harsh or extreme environments (Wei et al., 2020). *Lactobacillus*, a genus of
Gram-positive bacteria, has also been identified as abundant in atmospheric dust
(Federici et al., 2018; Xu et al., 2017). *Sphingomonas*, typically found in water bodies,
soil, and roots, can thrive in extreme environments (Hu et al., 2007; Sun et al., 2018).
It is usually more abundant in deserts and can be transported over long distances via air
masses. In contrast, *Comamonas* was identified as an indicator bacterium in marine air-
mass samples, which is dominant in coastal cities (Wei et al., 2020) and originates from



soil, activated sludge, and water (Yan et al., 2012).

For fungal community, *Aspergillus* (p=0.014) and *Malassezia* (p=0.041) were

significantly differentiated in different air masses. *Aspergillus* was 53.7% and 20.1% in
marine and terrestrial air-mass samples, respectively. *Aspergillus* is a dominant fungus
in offshore areas such as Qingdao, China (Li et al., 2011). Moreover, the Saprophytic
*Aspergillus* was also prevalent in clean samples during haze pollution episode and was
commonly detected on non-Haze days (Yan et al., 2016). Prior research has established
that *Aspergillus* is ubiquitously found in nature and non-polluted environments (Li and
Kendrick, 1995). *Malassezia* was higher in terrestrial and mixed air-mass samples,
which has been found to be widespread in a variety of animals. As a parasitic fungus,
*Malassezia* causes the majority of skin diseases, such as dandruff and seborrheic
dermatitis caused by *Malassezia sphericalis* (DeAngelis et al., 2007). The
*Cladosporium* is significantly higher in terrestrial and mixed air-mass samples than
marine air-mass samples, which is ubiquitous worldwide, commonly found in a wide
variety of plants and is frequently isolated from soils, paints, textiles, foodstuffs and
organic matters (Bensch et al., 2012; Ellis, 1977), known to be a common endophyte as
well as a foliar fungus (El-Morsy, 2000; Islam and Hasin, 2000). *Cladosporium* have
been commonly observed in terrestrial atmospheric environments, demonstrating the
potential origins from continental environments (Frączek et al., 2017; Han et al., 2019).





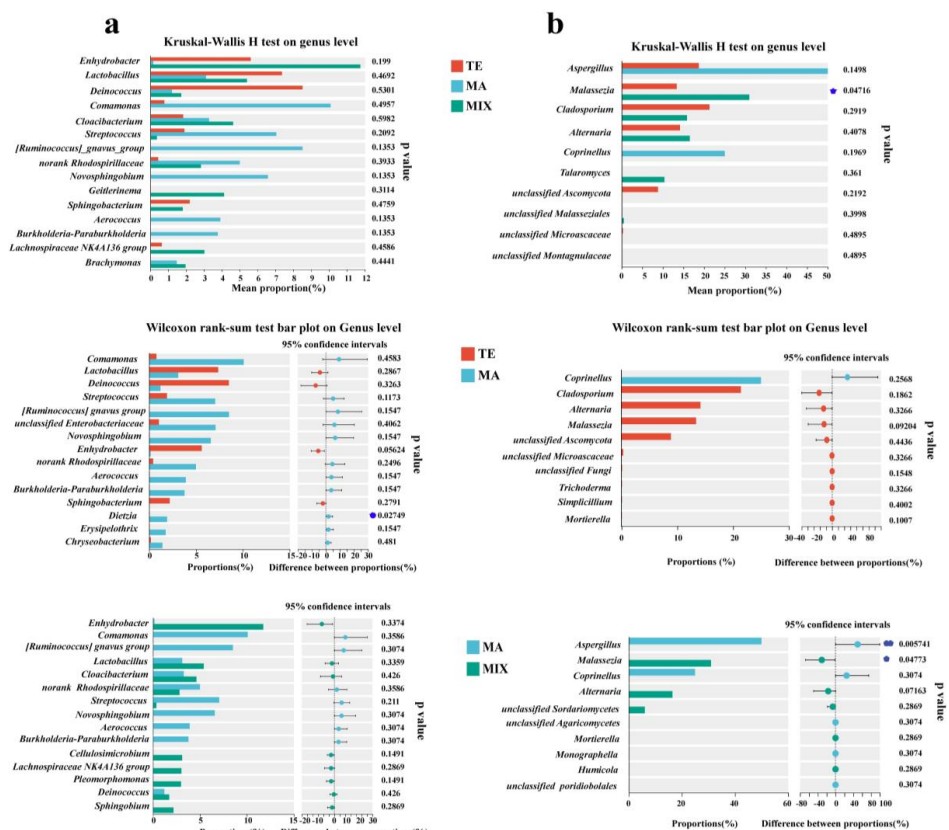

**415**

**416 Fig. 3 Bacterial (a) and fungal (b) community composition disparities influenced by different air**

**417 masses**

The FAPROTAX analysis revealed that the primary bacterial ecological functions
were chemoheterotrophy, aerobic chemoheterotrophy, fermentation, and human
pathogens. These accounted for 74%, 47%, and 44% respectively in terrestrial, marine,
and mixed air-mass samples. Notably, human pathogens and animal parasites or
symbionts were more prevalent in terrestrial and mixed air-mass samples than in marine
air-mass samples. Conversely, marine air-mass samples were enriched with mammal gut
bacteria, as well as hydrocarbon and automatic compound degradation bacteria. Overall,
the dominant airborne bacteria in the coastal city during winter primarily inhabited
anthropogenic environments such as soil, water, and terrestrial ecosystems. Additionally,
marine ecosystems served as a significant source of airborne microbes (Griffin, 2003;
Xu et al., 2019).

The fungal community function in terrestrial and mixed air-mass samples were



similar, with undefined Saprotroph, Animal Pathogen-Undefined Saprotroph, Animal
Pathogen-Endophyte-Lichen Parasite, Animal Pathogen-Endophyte-Plant Pathogen
were the main functions, which totaled 99.27%, 99.98%, 99.27% in the terrestrial,
marine, and mixed air-mass samples. In particular, the prevalence of Saprotroph fungi
was observed higher in samples from marine air masses, such as those containing
*Aspergillus* (Xu et al., 2017). Notably, the fungi associated with terrestrial air masses
predominantly carried animal pathogens and exhibited greater pathogenicity, including
species such as *Malassezia* and *Alternaria* (Gandolfi et al., 2013; Masiol et al., 2012).





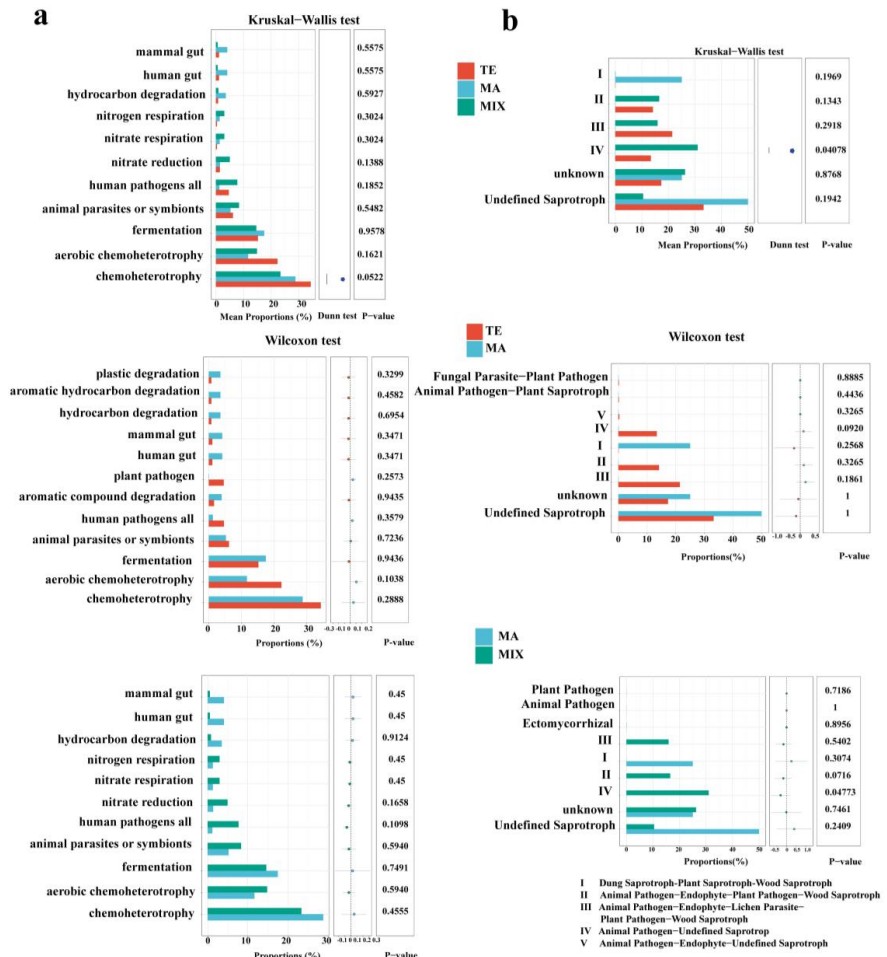

**Fig. 4. Bacterial (a) and fungal (b) community function disparities influenced by different air masses**



**3.5 Implications of environmental factors on coastal airborne microbes**


To understand the contribution of marine and terrestrial air masses to microbial
aerosols in the coastal city, the Mantel correlation analysis between microbial aerosols
and environmental factors were performed (Fig.5, Fig.S4). Influenced by terrestrial and
mixed air masses, bacterial community was significantly positively correlated with
$SO_4^{2-}$ ($P<0.05$), and bacterial and fungal communities were positively correlated with
continental ions such as $K^+$, $Mg^{2+}$, and $Ca^{2+}$ (Shi et al., 2022). Moreover, bacterial and
fungal concertation was positively correlated with $NO_2$ ($P < 0.05$) and significantly
negatively correlated with $PM_{10}$ (P<0.01).
Air parcels transported over long distances appear to harbor diverse microbial
populations (DeLeon-Rodriguez et al., 2013; Kakikawa et al., 2009). The long-distance
transportation of dust particles from the northwestern winds in the Inner Mongolia
region may have changed the community structure and abundance (Castañer et al., 2017;
Squizzato and Masiol, 2015). Dust-borne bacteria (*Staphylococcus*, *Delftia*,
*Pseudoalteromonas* and *Deinococcus*) were injected into the atmosphere during dust
events, and most of them accompanied the dust transportation to the downwind of Asian
Dust including the coastal city of Weihai. Influenced by the mixed air masses, bacterial
community was significantly positively correlated with $K^+$ ($P<0.01$) and negatively
correlated with $PM_{10}$ ($P<0.05$). Similarly, microbial communities showed high
positively correlated with ions from continental sources, such as $K^+$, $Mg^{2+}$, and $Ca^{2+}$,
which indicated that microorganisms carried by mixed air masses were mostly from
continental sources (Bates et al., 1992). A negative correlation was observed between
microbial communities and wind directions influenced by terrestrial and mixed air
masses. Wind blowing from continent or marine may play important role in microbial
community diversity (Jones and Harrison, 2004). Moreover, influenced by mixed air





mass, temperature have a greater impact on fungal community, which was positively
correlated with *Malasseziales* and *Davidiellaceae*.

The marine air masses are generally clean and have a strong scavenging effect on air

pollutants. Notably, bacterial community had high correlation with sea salt ions such as
$Mg^{2+}$, and certain fungi species such as *Talaromyces*, *Monographella* and *Phoma* were
correlated with $Na^+$ (*P < 0.05*), which suggesting an obvious influencing form marine
sources. A positive correlation between air temperature and certain microorganisms
(*Aerococcus*, *Cloacibacterium*, *Sphingobium*, *Enhydrobacterium*, *Davidiellaceae*,
*Malasseziales*) indicates that the increase in air temperature in spring favors the survival
of airborne microbes, especially for pathogenic bacteria.



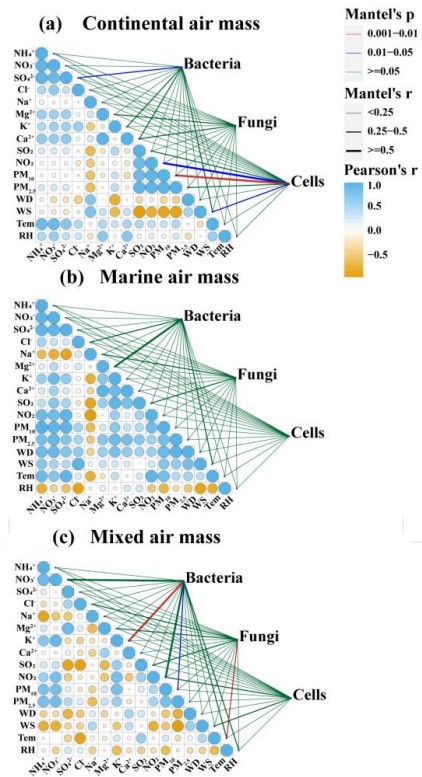

**Fig.5 Relationships between microorganisms and environmental factors influenced by different air masses, terrestrial air masses (a) marine air masses (b) and mixed air masses (c)**

This work shows that enormous levels of haze aerosols from the terrestrial and mixed air mass can be rapidly transported into the coastal city during regional haze pollution. During the long-distance transportation of air masses, a comprehensive mixture of soil-derived, biogenic, and anthropogenic microorganisms from terrestrial air masses, and aquatic, saprophytic, gut microorganisms from marine environments are fully mixed. This mixing is particularly evident in the samples of mixed air masses. Among that, not only large numbers of chemical components but also bacteria and fungi, as well as opportunistic pathogens, were transported into the coastal city. Microbial communities were strongly correlated with haze aerosols, e.g., WSIIs in $PM_{2.5}$ from terrestrial and



mixed air mass. The primary influence on terrestrial air mass was anthropogenic
emissions, with coal combustion for winter heating and biomass burning being the
predominant pollution sources. Moreover, dust events in spring carried higher
concentrations of particulate matter. These air pollutants can act as initial source of
bioaerosols such as bacteria and fungi, thereby providing a site for attachment
reproduction (Jiang et al., 2022). Additionally, water-soluble ions in PM, primarily
secondary ions, sulfate, nitrate, and ammonium ions, can supply essential nutrients for
microbial growth (Fan et al., 2019). This explanation elucidates the increased
concentration of particulate matter and microorganisms in terrestrial air mass during
heavy pollution. Simultaneously, it is pertinent to highlight that during pollution
incidents, the terrestrial air mass intensifies the pollution process. This intensification
results in a significant increase in the proportion of pathogenic microorganisms.
Conversely, the marine air mass facilitates the removal of pollution, introducing a
higher number of saprophytic microorganisms (Fig. 6). The presence of
microorganisms in marine air masses exhibited a significant correlation with sea salt
ions, specifically $Na^+$ and $Mg^{2+}$. These ions, when introduced into the atmosphere, form
aerosolized particulate matter. Additionally, meteorological factors exerted a more
pronounced influence on the concentrations of pollutants within marine air-mass
samples. Our research suggests that populations in coastal cities may also be susceptible
to exposure to these bioaerosols and pathogens, which are transported over long
distances during regional haze events.





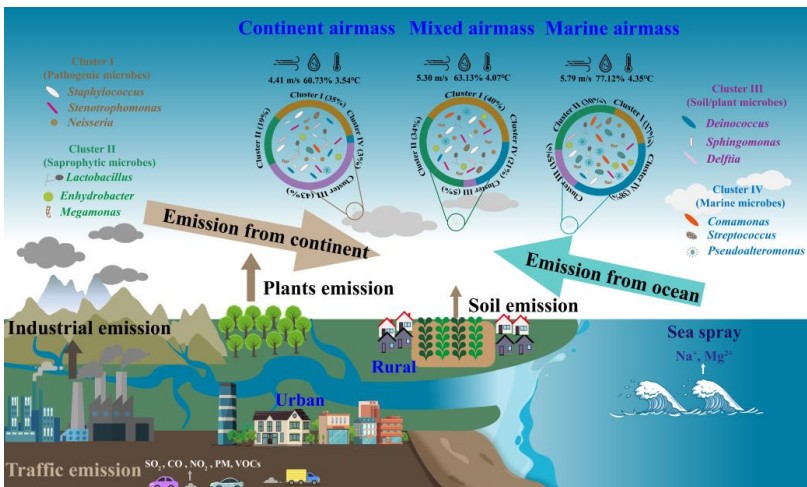

**Fig. 6 Implications of sea-land differences and various environmental factors, such as air pollutants and meteorological conditions, on the composition of airborne microbial communities. This is influenced by a range of air mass transmissions, including terrestrial, marine, and mixed air masses.**

## 4. Conclusion

This study evaluated the potential influence and the mixing effects taking place between natural and anthropogenic aerosols respectively from terrestrial and marine air-mass on biological constituents of PM$_{2.5}$ over coastal atmosphere. The concentration of water-soluble ions, metal elements and bioaerosols were higher in terrestrial and mixed air-mass samples than marine air-mass samples. However, the V/Ni ration greater than 0.7 in marine air-mass samples indicated significant influence from marine ship emissions. Bacterial and fungal community influenced by terrestrial and mixed air masses were enriched in animal pathogens, including the *Staphylococcu*, *Comamonas*, *Malassezia* and *Alternaria.* In contrast, marine air mass carried more saprophytic microbes, such as *Ruminococcus*, *Enterobacteriaceae*, *Aspergillus,* and *Davidiellaceae.* Potential implications of environmental factors on airborne microbes based on Mantel correlation analysis suggested that the bacterial community was mainly significantly



correlated with $SO_4^{2-}$, $K^+$, $Mg^{2+}$, and $Ca^{2+}$ and $PM_{10}$ influenced by terrestrial and mixed
air masses, which were mainly related to continental sources. In contrast, influenced by
marine air masses, bacterial and fungal community were strongly correlated with sea
salt ions, such as $Na^+$ and $Mg^{2+}$. The effects of meteorological factors on bioaerosols
vary significantly under different dominant air masses. The microbial communities
were found to be negatively correlated with wind direction, influenced by both
terrestrial and mixed air masses. Conversely, the predominant bacteria and fungi in
marine air-mass samples exhibited a positive correlation with air temperature,
suggesting elevated temperature in spring is conducive to the survival and reproduction
of bioaerosols. The present study demonstrated that air masses from different sources
would lead to different water-soluble ions, metal elements and microbial components
and concentrations in $PM_{2.5}$. This understanding will enhance our comprehension of the
environmental and climatic impacts on microbial aerosols within the marine boundary
layer. Our research offers a novel perspective on the variability of airborne microbial
communities and provides evidence suggesting that the atmospheric microbiome in
coastal cities is influenced by terrestrial and marine air masses. Further research is
required to conduct comprehensive studies on the sources, occurrence, and complex
amalgamation processes of coastal bioaerosols.
**CRediT authorship contribution statement**

All authors contributed to the manuscript and have given approval of the final version.

Min Wei designed the study. Qun He performed the data analysis and wrote the original
manuscript. Min Wei assisted with the sampling, Qun He, Zhaowen Wang and Rongbao
Duan conducted the experiments and performed the statistical analyses. Houfeng Liu,
Min Wei, and Pengju Xu contributed to the interpretation of results. Min Wei, Caihong



Xu and Jianmin Chen revised the manuscript.

**Data availability**

Hourly and daily average air quality and meteorological factor data were obtained
from Shandong Province Ecological Environment Big Data Platform
(http://27.211.168.253:18102/portal/); MeteoInfo backward trajectory model
(MeteoInfo 3.7.4 - Java, http://www.meteothink.org /downloads/index.html);
meteorological data were obtained from GDAS1
(ftp://arlftp.arlhq.noaa.gov/pub/archives/gdas1/), and meteorological data as well as
bacterial and fungal concentrations are available from the authors on request
(minwei@sdnu.edu.cn). The original 16S rRNA and ITS gene sequences are available
in the Sequence Read Archive (SRA) under accession number PRJNA1096829.

**Declaration of competing interest**

The authors declare that they have no conflict of interests.

**Acknowledgments**

This research was supported by the National Key Research and Development
Program of China (2023YFC3710200), National Natural Science Foundation of China
(42075183), and China Postdoctoral Science Foundation (2019T120606).

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
