# Peer review of "intermixing of bioaerosols over coastal atmosphere"

_EGUsphere, 2024_

## Author Comment (AC1)

**Influence of terrestrial and marine air-mass on the constituents and intermixing of bioaerosols over coastal atmosphere**

Qun He[a], Zhaowen Wang[a], Houfeng Liu[a], Pengju Xu[a], Rongbao Duan [a], Caihong Xu[b],

Jianmin Chen[b], Min Wei[a,b],*

[a] College of Geography and Environment, Shandong Normal University, Ji'nan 250014, China

[b] Shanghai Key Laboratory of Atmospheric Particle Pollution and Prevention (LAP³), Fudan

Tyndall Centre, Department of Environmental Science & Engineering, Fudan University,

Shanghai 200433, China

* Correspondence to: Min Wei (minwei@sdnu.edu.cn)

**Response to editor**

We thank the editor for the opportunity to revise our manuscript, we appreciate the editor and reviewers very much for their positive and constructive comments and suggestions. In the attachment, we have responded to the reviewer's comments respectively. We also revised our manuscript according to the reviewer's comments and attach a revised manuscript with tracked changes later. The amendments were marked in red in the revised manuscript, which we would like to submit for your kind consideration.

**Response to reviewer 1**

In the study titled 'Impact Assessment of Terrestrial and Marine Air Mass on the Constituents and Intermixing of Bioaerosols over Coastal Atmosphere,' the authors aim to explore the effects of sea-land air mass exchange on the spatial and temporal distribution of aerosols, as well as potential intermixing processes in coastal regions.

In my opinion, the study demonstrates considerable potential: it integrates current molecular biology methods applied to atmospheric aerosol sampling at low altitude (15 meters above the ground) with concepts of atmospheric physics to understand the effects of mixing at the land-ocean interface on the ion and trace metal composition of air masses and their associated microbial composition (bacteria and fungi). The study is comprehensive, relying on current methods for both sample processing and statistical analysis. The results are interesting, original, and relevant to a field that remains under-researched.

However, I find the paper long and often overly descriptive. The message is not always clear, particularly regarding the microbial communities section, with some inaccuracies present. Some parts are also too speculative, in my opinion, and not sufficiently connected to the current literature. Overall, I believe the results and discussion sections need thorough revision

That said, I believe this article deserves to be published in EGUsphere, provided certain major changes are made.

**Response of the authors:**

We greatly appreciate the positive comments and constructive suggestions on our manuscript. According to your suggestions, we have streamlined the results and discussion sections of the article, made detailed revision based on your comments, and incorporated them into the revised manuscript.

General comments:

1. Field blanks are missing, raising the question of how to evaluate the effect of cross-contamination between air masses. This issue is crucial and must be addressed, as air samples are highly susceptible to contamination due to their low microbial biomass.

**Response of the authors:** Thank you for the suggestion.

During the sampling process, blank samples are concurrently gathered with test samples. These blank samples were procured by inserting the membrane into the sampler head

without initiating the sampler. After a duration of 23 hours, the blank samples were extracted simultaneously with the collected sample membrane. They were then placed in a sterile filter cassette, encased in clean aluminum foil, and stored at -80°C until further analysis.

Air masses were categorized based on their dominant characteristics. Air masses constituting more than 90% of a given day were classified as land air masses originating from the land, while those exceeding 90% were identified as ocean air masses originating from the ocean. Parallel samples were established for both land and ocean air masses. Additionally, a blank control was implemented to assess potential cross-contamination between these air masses.

2. I am also skeptical about the calculation of 24-hour back trajectories. I believe there is a considerable loss of information that may affect the interpretation of the data. The region in question is influenced by multiple geographical conditions. Extending the backward trajectories could provide better insights into the patterns of microbial assemblages in the observed air masses, especially in the context of long-distance transport

**Response of the authors:** The selection of the 24-hour backward trajectory is predicated on the sampling time. The sampling time was, with samples gathered twice daily (7:00~18:30 and 19:00~06:30 the following day). Consequently, we opted for a 24-hour backward trajectory to more distinctly analyze the impact of the air mass on each sample's results.

Additionally, we calculated both 48-hour and extended (72-hour) backward trajectories. The direction of the primary air mass source remained approximately consistent with the 24-hour backward trajectories (Figure 1).

[Figure]

**Figure 1 Typical 24h (a), 48h (b) and 72h (c) backward trajectories during the sampling period.**

3. How can the local contribution to pollution be determined? The authors state that 60% of their air masses have a terrestrial origin, yet only three episodes of severe pollution were observed. How can they be certain that this pollution originates from distant sources?

**Response of the authors:** The three pollution events were characterized by regional air pollution. During these pollution episodes, both the air mass backward trajectories and satellite images revealed that the entire northern region of China, spanning from the North China Plain to the coastal areas, suffered severe air pollution.

The percentage of terrestrial air masses during the sampling period is presented in Table S1. The backward trajectories at 24 h and 72 h indicate the main sources of the polluted air masses (Figure. 2). The main sources of air masses during air pollution process were from Northwestern Inner Mongolia and the Southwestern inland areas. Furthermore, satellite images from National Aeronautics and Space Administration (http://worldview.earthdata.nasa.gov) indicated the long-range transport of aerosol particles. For instance, in pollution episode 2, which was haze-dust composite pollution, the polluted air mass from Inner Mongolia increased the concentration of $PM_{10}$ due to the influence of the dust event. Meanwhile, air pollutants emitted from coal

combination in the heating season in northern China increased in March. Under unfavorable meteorological conditions such as high humidity, low air pressure, and low wind speeds, the diffusion of air pollutants was impeded which led to the intensification of the pollutants' accumulation, and the whole North China Plain to the eastern coast was covered by haze particles.

[Figure]

**Figure 2 Backward trajectories and satellite images of the three pollution episodes**

**Table S1 Sample classification and description influenced by different air masses**

| Group | | | | | | | | | | | |
|-------|---|---|---|---|---|---|---|---|---|---|---|
| | **Samples** | | | | | | | | | | |
| | Date | 19-Jan | 20-Jan | 1-Mar | 9-Mar | 11-Mar | 12-Mar | 22-Mar | 23-Mar | 24-Mar | 25-Mar |
| **TE** | $PM_{2.5}$ | 51.35 | 124.67 | 30.58 | 16.5 | 65.63 | 38.21 | 53.05 | 77.46 | 84.42 | 60.83 |
| | Sample | WH_1 | WH_2 | WH_7 | WH_8 | WH_10 | WH_11 | WH_15 | WH_16 | WH_17 | WH_18 |
| | Date | 21-Jan | 16-Mar | 21-Mar | 29-Mar | 30-Mar | 31-Mar | | | | |
| **MA** | $PM_{2.5}$ | 7.92 | 12.38 | 15.37 | 25.38 | 25.42 | 28.83 | | | | |
| | Sample | WH_3 | WH_13 | WH_14 | WH_22 | WH_23 | WH_24 | | | | |
| | Date | 26-Feb | 27-Feb | 28-Feb | 10-Mar | 14-Mar | 26-Mar | 27-Mar | 28-Mar | | |
| **MIX** | $PM_{2.5}$ | 41.12 | 60.61 | 41 | 51.63 | 36.42 | 52.58 | 64.69 | 48.58 | | |
| | Sample | WH_4 | WH_5 | WH_6 | WH_9 | WH_12 | WH_19 | WH_20 | WH_21 | | |
| **$PM_{2.5}$ concentration (μg/m$^3$)** | <35 | 35-75 | 75-150 | 150-250 | >250 | | | | | | |

TE, terrestrial air mass; MA, marine air mass; MIX, mix air mass.

4. Specific Comments:

Title: The word "impact" implies an environmental impact study or risk assessment, which is not the case. I suggest modifying it slightly to better reflect the characterization aspect.

**Response of the authors:** Thank you for your suggestion, we have revised the title.

*Influence of terrestrial and marine air-mass on the constituents and intermixing of bioaerosols over coastal atmosphere*

5. L125-127 and 163-164: How do the authors link air masses at an altitude of 500 meters with their sampling site at 15 meters?

**Response of the authors:** The air mass may sink during transmission, with the 500 m air mass typically influencing the near-surface (Shrestha et al., 2023; Kawana et al., 2021; Xu et al., 2022). Meanwhile, our sampling referred to the requirements of the national standard HJ/T194, the entrance of the sampler is about 15 m from the ground height when sampling, and the altitude of 500 m represents the long-distance transmission compared with other heights of the air mass can better show the source of the air mass (Lang-Yona et al., 2020; Wani et al., 2021) as shown in Figure 3. If the altitude is chosen too low, it will be more affected by local anthropogenic pollution sources near the ground.

[Figure]

**Figure 3 Backward trajectories at different altitudes, 50 m (a), 500 m(b), 1000 m(c).**

*Reference*

*Kawana, K., Matsumoto, K., Taketani, F., Miyakawa, T., and Kanaya, Y.: Fluorescent biological aerosol particles over the central Pacific Ocean: covariation with ocean surface biological activity indicators, Atmos. Chem. Phys., 21, 15969-15983, 10.5194/acp-21-15969-2021, 2021.*

*Shrestha, S., Zhou, S., Mehra, M., Guagenti, M., Yoon, S., Alvarez, S. L., Guo, F., Chao, C.-Y., Flynn Iii, J. H., Wang, Y., Griffin, R. J., Usenko, S., and Sheesley, R. J.: Evaluation of aerosol- and gas-phase tracers for identification of transported biomass burning emissions in an industrially influenced location in Texas, USA, Atmos. Chem. Phys., 23, 10845-10867, 10.5194/acp-23-10845-2023, 2023.*

*Xu, X., Xie, J., Li, Y., Miao, S., and Fan, S.: Measurement report: Vehicle-based multi-lidar observational study of the effect of meteorological elements on the three-dimensional distribution of particles in the western Guangdong–Hong Kong–Macao Greater Bay Area, Atmos. Chem. Phys., 22, 139-153, 10.5194/acp-22-139-2022, 2022.*

*Lang-Yona, N., Öztürk, F., Gat, D., Aktürk, M., Dikmen, E., Zarmpas, P., Tsagkaraki, M., Mihalopoulos, N., Birgül, A., Kurt-Karakuş, P. B., and Rudich, Y.: Links between airborne microbiome, meteorology, and chemical composition in northwestern Turkey, Sci. Total Environ., 725, 10.1016/j.scitotenv.2020.138227, 2020.*

*Wani, M. A., Mishra, A. K., Sharma, S., Mayer, I. A., and Ahmad, M.: Source profiling of air pollution and its association with acute respiratory infections in the Himalayan-bound region of India, Environ. Sci. Pollut. Res., 28, 68600-68614, 10.1007/s11356-021-15413-0, 2021.*

6. L129 – 131: The authors specified the use of $PM_{2.5}$ samplers equipped with 47 mm quartz filters but did not provide specific details about the type of $PM_{2.5}$ sampler used, such as the flow rate and manufacturer information. Please specify.

Additionally, did the authors use field blanks as controls? Was the device cleaned between quartz filter changes? If so, please specify the cleaning procedure. Furthermore, was the air flow rate calibrated before and after sampling to ensure accurate measurements?

**Response of the authors:** Thank you for your suggestions, we have added the detailed information in Line 125-144.

*Two parallel $PM_{2.5}$ samplers (TH‐150C‐III, Wuhan Tianhong Instrument Co., Ltd., China) with a particle size of 2.5 ± 0.2 μm were utilized equipped with Quartz membrane for the collection of $PM_{2.5}$, inorganic ion, metal elements, and*

*microorganisms at a flow rate of 100 L min$^{-1}$. The Quartz membranes were cauterized in a muffle furnace at 450°C for 6 hours to remove carbonaceous and contaminant materials. Intermittent sampling was used and samples were collected twice a day (7:00~18:30 and 19:00~06:30 the following day). During the sampling process, the pre-weighed filter membrane was carefully positioned onto the filter mesh within a clean sampling folder using tweezers. The membrane's rough side should be oriented towards the direction of air intake, and it should be firmly pressed until there is no detectable air leakage. Prior to membrane replacement, the membrane tray should be wiped with alcohol cotton and dried. After sampling, it should be removed using tweezers. Blank samples were procured by putting the membrane into the sampler head without initiating the sampler. After sampling, the blank samples were extracted simultaneously with the collected sample membrane. All samples were then placed in a sterile filter cassette, encased in clean aluminum foil, and stored at -80°C until further analysis. During the whole sampling period from January 1 to March 31, a total of 102 PM$_{2.5}$ samples over 51 days were obtained, out of which 24 samples were chosen for subsequent analysis based on air mass categories.*

7. L144-145: "The membranes were meticulously weighed using a Mettler XP-6 balance with an accuracy of 10$^{-6}$ g. Prior to weighing, the membranes were maintained in a controlled environment with consistent temperature and humidity for a duration of 24 hours."

I assume this step aims to remove water from the filters. How did the authors ensure that all residual water was removed, given that residual moisture can significantly affect the filters' weight? Additionally, did the authors sample during rainfall events? Is the sampler equipped with a rain cap? Were the filters exposed to rain, and if so, how were they processed?

**Response of the authors:** Maintaining a consistent temperature and humidity environment for the filter membrane both before and after sampling is crucial to ensure accurate weighing of PM$_{2.5}$ on the membrane and determination of water-soluble ions

within it. The sampling process employs parallel sampling, with each sample analyzed separately for chemical and biological components. A stable temperature and humidity environment is essential for the accurate determination of chemical components. For the analysis of biological components, the filter membrane was collected, placed in a sterile filter box, wrapped in clean aluminum foil, and stored at -80°C until further examination.

Weather conditions were consistently monitored throughout the sampling process. Should snowfall or rainfall occur during the sampling period, it is imperative to cease filter membrane sampling on rainy or snowy days and safeguard the instrument by enveloping it with a waterproof cloth (Figure 4). Simultaneously, the sampler is equipped with its own anti-rain and snow settings, ensuring that minor rainfall has minimal impact on the sampler.

[Figure]

**Figure 4 The Parallel samplers and the snowfall during sampling (Jan 22)**

8. L149-155: The protocol for the characterization of ions and metallic elements lacks details. Please specify the sonication time, the type of column used, and the reference for the conductivity detector.

**Response of the authors:** Thanks to your suggestion, we have added the detailed information in Line 149-164.

*The mass concentration of PM$_{2.5}$, water-soluble ions, and metal elements were quantified after sampling. The membranes were meticulously weighed utilizing a Mettler XP-6 balance with an accuracy of 10$^{-6}$ g. Prior to the weighing, the membranes were maintained in a controlled environment with consistent temperature and humidity for a duration of 24 hours. Ion chromatography (ICS-2100, Chameleon 6. 8, AS-DV autosampler Thermo Fisher) was employed to determine the concentration of water-soluble ions such as Na$^+$, K$^+$, Ca$^{2+}$, Mg$^{2+}$, Cl$^-$, NO$_3^-$, SO$_4^{2-}$, and NH$_4^+$. These ions were extracted by sonication with ionized water for 45 min and then separated by anion (IonPacAS23) or cation column exchange (IonPacCS12A). Then, these ions were detected using a conductivity detector with an anion separation column of IonPacAS23, flow rate: 1.0 mL/min, an anion suppressor of AERS500, and conductivity detector. The injection volume was 25 μL, and the cation separation column was IonPacCS12A, flow rate:1.0 mL/min, cation suppressor was CERS500 (Zhang et al., 2022). Metallic elementals including Al, Fe, Ti, Mn, Co, Ni, Cu, Zn, Ga, Sr, Cd, Sn, Sb, Pb, V, Cr, and As, were extracted using microwave digestion extraction (ETHOS ONE, Milestone), with the concentrations determined by ICP-MS or ICP-OES (Thermo Fisher).*

9. L163-168: The authors defined their criteria for classifying air masses as continental or marine but did not explicitly mention the method they used to determine the origins of air masses. Please elaborate on the method used to categorize the air masses as marine or terrestrial.

**Response of the authors:** In order to avoid the subjective errors caused by purely relying on visual observation and experience, this study quantitatively determines the properties of the air masses arriving at the observatory every hour based on the ratio of the length of the trajectory of the air masses passing through the ocean area to the total length. This part is explained in the article, and the categorization method is as follows in Line 173-177.

*In this study, air mass categories were defined that if more than 90% of the masses originated from the ocean it was a typical sample affected by marine air-*

*mass; a typical terrestrial air-mass sample if more than 90% originated from the continent; and mixed air-mass sample if the proportions of terrestrial and marine air masses were similar or accounted for more than 40% of the total air masses in one day.*

10. L170-171: The authors state that they extracted DNA directly from the quartz filters. Did they use the same filters that were used for the previous ion and metallic element determinations? Additionally, was the DNA extracted from the entire filter or just a part of it?

**Response of the authors:** In this study, we conducted parallel sampling, wherein two parallel membranes were procured in a single sampling event. One membrane was used for PM$_{2.5}$ mass concentration, water-soluble ions, and metal element measurements, and stored at 4 °C prior to analysis. Another membrane for bioaerosol analysis was stored at -80 °C to avoid microbial degradation or growth prior to further analysis.

11. 172-173: The extracted DNA was measured using a Nanodrop spectrophotometer (Nanodrop 2000, Thermo Scientific USA) to determine the concentration. I am surprised that the authors were able to quantify the DNA in their samples using a Nanodrop, considering its sensitivity is around 2 ng/µl. Could the authors provide more details about the measured DNA concentrations to clarify?

**Response of the authors:** We reviewed relevant literature about the measurement of DNA concentration via Nanodrop (Nanodrop 2000, Thermo Scientific USA) (Bai et al. 2020; Shi et al. 2022; Xu et al. 2020; Zhong et al. 2019). The DNA concentration in the collected samples ranged from 10 to 20 ng/µL, thereby satisfying the prerequisites for subsequent PCR amplification.

*Reference*

*Bai, W., Li, Y., Xie, W., Ma, T., Hou, J., and Zeng, X.: Vertical variations in the concentration and community structure of airborne microbes in PM$_{2.5}$, Sci. Total Environ., 760, 10.1016/j.scitotenv.2020.143396, 2021*

*Shi, Y., Lai, S., Liu, Y., Gromov, S., and Zhang, Y.: Fungal Aerosol Diversity Over the Northern South*

*China Sea: The Influence of Land and Ocean, J. Geophys. Res.: Atmos., 127, 10.1029/2021jd035213, 2022.*

*Xu, C., Chen, J., Wang, Z., Chen, H., Feng, H., Wang, L., Xie, Y., Wang, Z., Ye, X., Kan, H., Zhao, Z., and Mellouki, A.: Diverse bacterial populations of $PM_{2.5}$ in urban and suburb Shanghai, China, Front. Environ. Sci. Eng., 15, 10.1007/s11783-020-1329-7, 2020.*

*Zhong, S., Zhang, L., Jiang, X., and Gao, P.: Comparison of chemical composition and airborne bacterial community structure in $PM_{2.5}$ during haze and non-haze days in the winter in Guilin, China, Sci. Total Environ., 655, 202-210, 10.1016/j.scitotenv.2018.11.268, 2019.*

12. L234-238: The phrasing of this sentence is confusing. It appears to mix the concepts of Mantel correlation analysis and Spearman's rank correlation coefficient. Could the authors clarify if the Mantel test was conducted using Spearman's rank correlation coefficient as the measure of correlation, or if separate Spearman's rank correlation analyses were performed? A clear explanation of the statistical methods used would be helpful.

**Response of the authors:** We have revised the sentence in Line 245-248.

*The Mantel analysis was utilized to reveal the correlation between microbial community and various environmental factors. The Pearson's r coefficient at p<0.05 and p<0.01 indicates the significant correlation.*

13. 275 :277: "The introduction of marine air mass from the west led to the elimination of the pollution, resulting in an average $PM_{2.5}$ concentration reduction to 7.92 μg /m³." I understand that dilution could be a factor, but stating that there was an elimination of pollution seems uncertain. Did the air masses in the following days also have a terrestrial origin? Please elaborate.

**Response of the authors:** Pollution episode I occurred from January 19 to January 23, 2018 was typical haze pollution. Winter in northern China is characterized by a high incidence of haze pollution, specifically in January and February. Under the influence of the northern Siberian cold high-pressure system, pollution events are predominantly

associated with northwesterly cold air currents, with the haze pollution air mass originating from the north and northwest regions. The backward trajectory shows that the air mass originated from land during the pollution elimination phase (Jan. 22nd and 23rd) (Figure 5). Meanwhile, meteorological data indicate the removal of pollution by snowfall processes, high winds and the cold northwestern air brought snowfall process, high winds, and cold high pressure on the pollution removal, and pollutant concentrations decreased (Figure 6).

[Figure]

**Figure 5 Backward trajectory diagram of pollution episode I.**

[Figure]

**Figure 6 Meteorological factors and pollution processes during pollution episode I**

14. L290-293: "The concentrations of $NO_3^-$, $SO_4^{2-}$, and $NH_4^+$ were significantly influenced by both terrestrial and mixed air masses, with the latter exhibiting a more pronounced effect ($NO_3^-$, 10.65±3.26 μg/m³; $NH_4^+$,7.39±3.30 μg/m³; $SO_4^{2-}$ 6.76±1.77 μg/m³)"

I do not see any statistical test mentioned, and the standard errors (if they are standard errors, as this is not specified in the legend) seem large. Furthermore, there are no standard errors reported for the trace metal elements. Why is this? Please revise.

**Response of the authors:** Thank you for your suggestion, we have added the standard errors for the trace metal elements in Fig. S3. The terrestrial air masses include the clean samples and polluted samples (Table S1), which resulted in large standard errors.

[Figure]

**Fig. S3 Water-soluble ions (a) and metal element concentration (b, c) in PM₂.₅ influenced by terrestrial, marine and mix air mass.**

15. L327-328: Unless I am mistaken, the authors did not measure the bacterial and fungal concentrations directly but estimated their concentrations based on the quantity of 16S rRNA genes, which is not exactly the same. Additionally, do these concentrations apply to all air masses combined? Please revise.

**Response of the authors:** In the present study, the bacterial and fungal concentrations were estimated based on the quantity of 16S rRNA genes. These concentrations represent the average of all samples during the sampling period. Samples collected affected by terrestrial air masses were screened out, and the microbial concentration

obtained from these samples were analyzed and processed together. The mean and standard deviation were calculated, which is the microbial concentration of terrestrial air masses, and the microbial concentration of marine air masses and mixed air masses were obtained by the same method.

The qPCR method has been widely used in airborne microbial concentration investigation. Jiang et al. (2022) comprehensive analyzed the microbial community structure in the air of Hefei City during different seasons using qPCR and high-throughput sequencing. Rodríguez et al. (2020) used an ABI PRISM 7500 Rapid Sequence Detection System (Applied Biosystems, Foster City, CA. USA) to perform three quantitative PCRs (qPCRs) and to estimate airborne abundance of bacteria, and the abundance of airborne fungi was estimated by quantifying 18S rRNA gene copy number. Zhen et al. (2017) estimated the abundance of airborne bacteria in Beijing by quantifying the copy number of 16S rRNA gene. In most of these studies, bacteria were identified by 16S rRNA sequencing (Hu et al., 2018).

Microbial concentrations can also be determined using culture-dependent method (Bowers et al., 2012) or fluorescent staining (Yin et al., 2021). The culture-dependent method targeted cultured colonies rather than total bacterial or fungal species. Fluorescent staining identifies microbial cells by fluorescent color, size, and shape. From the determined bacterial abundance and size, the biovolume and biomass of bacteria in the atmosphere can be similarly estimated. However, the interference of non-biological components is relatively high.

*Reference*

*Bowers, R. M., McCubbin, I. B., Hallar, A. G., and Fierer, N.: Seasonal variability in airborne bacterial communities at a high-elevation site, Atmos. Environ., 50, 41-49, 10.1016/j.atmosenv.2012.01.005, 2012.*

*Hu, W., Niu, H., Murata, K., Wu, Z., Hu, M., Kojima, T., and Zhang, D.: Bacteria in atmospheric waters: Detection, characteristics and implications, Atmos. Environ., 179, 201-221, 10.1016/j.atmosenv.2018.02.026, 2018.*

*Jiang, S., Sun, B., Zhu, R., Che, C., Ma, D., Wang, R., and Dai, H.: Airborne microbial community*

*structure and potential pathogen identification across the PM size fractions and seasons in the urban atmosphere, Sci. Total Environ., 831, 10.1016/j.scitotenv.2022.154665, 2022.*

*Rodríguez, A., Seseña, S., Sánchez, E., Rodríguez, M., Palop, M. L., Rodríguez Martín-Doimeadios, R. d. C., and Rodríguez Fariñas, N.: Temporal variability measurements of PM2.5 and its associated metals and microorganisms on a suburban atmosphere in the central Iberian Peninsula, Environ. Res., 191, 10.1016/j.envres.2020.110220, 2020.*

*Yin, Y., Qi, J., Gong, J., and Gao, D.: Distribution of bacterial concentration and viability in atmospheric aerosols under various weather conditions in the coastal region of China, Sci. Total Environ., 795, 10.1016/j.scitotenv.2021.148713, 2021.*

*Zhen, Q., Deng, Y., Wang, Y., Wang, X., Zhang, H., Sun, X., and Ouyang, Z.: Meteorological factors had more impact on airborne bacterial communities than air pollutants, Sci. Total Environ., 601-602, 703-712, 10.1016/j.scitotenv.2017.05.049, 2017.*

16. L332-333: "The terrestrial air masses came from the inland areas, which carried more microorganisms from anthropogenic activities and natural sources with high resistance to high temperatures, dryness, and strong ultraviolet rays (Gong et al., 2020)". The current phrasing of this sentence is odd. The paragraph that discusses the different genera and their resistance to temperature, dryness, and UV radiation appears later (L348). As it stands, this sentence seems more speculative.

**Response of the authors:** We have checked and added relevant literatures in Line 401-408.

*Bioaerosols originating from anthropogenic environments, such as sewage treatment plants and hospitals, may harbor more resistant bacteria. Microorganisms from arid areas like deserts possess a higher number of species adapted to harsh conditions, including resistance to temperature fluctuations, dryness, and UV radiation. These include bacteria such as Bacillus, Streptococcus, and Deinococcus (Maki et al., 2010; Park et al., 2018; Qi et al., 2021). These microorganisms can adhere to particulate matter during long-distance transport of airmasses after being released from their terrestrial habitats.*

*Reference*

*Maki, T., Susuki, S., Kobayashi, F., Kakikawa, M., Tobo, Y., Yamada, M., Higashi, T., Matsuki, A., Hong, C., Hasegawa, H., and Iwasaka, Y.: Phylogenetic analysis of atmospheric halotolerant bacterial communities at high altitude in an Asian dust (KOSA) arrival region, Suzu City, Sci. Total Environ., 408, 4556-4562, 10.1016/j.scitotenv.2010.04.002, 2010.*

*Park, J., Li, P.-F., Ichijo, T., Nasu, M., and Yamaguchi, N.: Effects of Asian dust events on atmospheric bacterial communities at different distances downwind of the source region, J. Environ. Sci., 72, 133-139, 10.1016/j.jes.2017.12.019, 2018.*

*Qi, J., Huang, Z., Maki, T., Kang, S., Guo, J., Liu, K., and Liu, Y.: Airborne bacterial communities over the Tibetan and Mongolian Plateaus: variations and their possible sources, Atmos. Res., 247, 10.1016/j.atmosres.2020.105215, 2021.*

17. L333-335: Please specify the statistical test used and the associated p-value in the text.

**Response of the authors:** We have specified the statistical tests and added the associated p-value in the revised manuscript in Line 338-347.

Based on the 24-h backward trajectories from the software of MeteoInfo (MeteoInfo 3.7.4 - Java, http://www.meteothink.org/downloads/index.html), samples were categorized into terrestrial, marine and mixed samples. All samples affected by terrestrial air masses were analyzed together, including clean and polluted samples. The mean and standard deviation were then calculated, which is the microbial concentration affected by terrestrial air masses. Microbial concentration of marine air masses and mixed air masses were obtained by the same method.

*Airborne bacterial and fungal concentration in $PM_{2.5}$ were $1.99\pm1.46\times10^5$ cells/$m^3$ (P=0.06) and $3.39\pm1.10\times10^4$ cells/$m^3$ (P=0.04), respectively. A high concentration was observed in terrestrial air-mass samples, with the average value of $4.72\pm3.93\times10^5$ cells/$m^3$ (P=0.12) and $6.37\pm1.70\times10^4$ cells/$m^3$ (P=0.007) for bacteria and fungi, respectively. Microbial concentration in marine air-mass samples was significantly lower, with the average concentrations of bacteria and fungi being $4.91\pm1.82\times10^4$*

*cells/m³ (P=0.04) and 6.15±3.09×10³ cells/m³ (P=0.07).*

18. 353-358: The authors claim that the higher abundance of *Cyanobacteria* originates from soils. However, to my knowledge, *Cyanobacteria* are mostly found in aquatic environments, although their presence in soils has been reported many times. Could this not indicate that the air masses had significant contact with marine surfaces? Additionally, why were the backward trajectories calculated over 24 hours and not 72 hours or more? A longer period would provide a better understanding of how the air masses are influenced by various surfaces.

**Response of the authors:** We have revised the description of *Cyanobacteria* is in Line 355-362.

*Cyanobacteria exhibit a higher concentration in both terrestrial and mixed air-mass samples. Cyanobacteria are prevalent in freshwater, soil, surface crusts in deserts (Cordeiro et al., 2020; Temraleeva et al., 2016; Curren and Leong, 2020). In comparison to terrestrial environments, the prevalence of Cyanobacteria is notably diminished in colder marine environments (Koh et al., 2012). Atmospheric Cyanobacteria in the coastal city may originate from the terrestrial environment, subsequently transported via terrestrial and mixed air masses.*

The selection of the 24-hour backward trajectory is predicated on the sampling time. The sampling time was, with samples gathered twice daily (7:00~18:30 and 19:00~06:30 the following day). Consequently, we opted for a 24-hour backward trajectory to more distinctly analyze the impact of the air mass on each sample's results. Additionally, we calculated both 48-hour and extended (72-hour) backward trajectories. The direction of the primary air mass source remained approximately consistent with the 24-hour backward trajectories (Figure 7).

[Figure]

**Figure 7 Typical 24h(a), 48h(b) and 72h(c) backward trajectories during the sampling period.**

*Reference*

*Cordeiro, R., Luz, R., Vasconcelos, V., Gonçalves, V., and Fonseca, A.: Cyanobacteria Phylogenetic Studies Reveal Evidence for Polyphyletic Genera from Thermal and Freshwater Habitats, Diversity, 12, 10.3390/d12080298, 2020.*

*Curren, E. and Leong, S. C. Y.: Natural and anthropogenic dispersal of cyanobacteria: a review, Hydrobiologia, 847, 2801-2822, 10.1007/s10750-020-04286-y, 2020.*

*Temraleeva, A. D., Dronova, S. A., Moskalenko, S. V., and Didovich, S. V.: Modern methods for isolation, purification, and cultivation of soil cyanobacteria, Microbiology, 85, 389-399, 10.1134/s0026261716040159, 2016.*

*Koh, E. Y., Cowie, R. O. M., Simpson, A. M., O'Toole, R., and Ryan, K. G.: The origin of cyanobacteria in Antarctic sea ice: marine or freshwater?, Environ. Microbiol. Rep., 4, 479-483, 10.1111/j.1758-2229.2012.00346.x, 2012*

19. L372-376: "*Streptococcus* are mostly found in the oral and gastrointestinal tracts of a variety of mammals and have not been shown to play a role in human infections to date (James et al., 2015). *Ruminococcus*, and unclassified Enterobacteriaceae are gut

microorganisms that may be related to the marine fish and other animal gut microbes."
The sentence is not correct. *Streptococcus* plays a significant role in human infections.
Additionally, *Enterobacteriaceae* are also found in the human microbiome. Please
rectify.

**Response of the authors:** Thank you for your suggestion, we have revised the sentence
in Line 436-441.

*Streptococcus are mostly found in the oral and gastrointestinal tracts and also play an*
*important role in human infections (Brouwer et al., 2023). Ruminococci, and*
*Enterobacteriaceae are typical gut microorganisms present in humans and animals (Ze*
*et al., 2012). In the coastal atmosphere, the identified sequences of these gut bacteria*
*might be associated with intestinal microorganisms in marine fish and mammal animals.*
*Reference*

*Brouwer, S., Rivera-Hernandez, T., Curren, B. F., Harbison-Price, N., De Oliveira, D. M. P.,*
*Jespersen, M. G., Davies, M. R., and Walker, M. J.: Pathogenesis, epidemiology and control of*
*Group A Streptococcus infection, Nat. Rev. Microbiol., 21, 431-447, 10.1038/s41579-023-00865-7,*
*2023.*

*Ze, X., Duncan, S. H., Louis, P., and Flint, H. J.: Ruminococcus bromii is a keystone species for the*
*degradation of resistant starch in the human colon, The ISME Journal, 6, 1535-1543,*
*10.1038/ismej.2012.4, 2012.*

20. L394-396: "In contrast, *Comamonas* was identified as an indicator bacterium in
marine air-mass samples, which is dominant in coastal cities (Wei et al., 2020) and
originates from soil, activated sludge, and water (Yan et al., 2012)."
I am unsure how to interpret this sentence. The authors identified *Comamonas* as a
marker of marine air mass, but they immediately suggest a soil/anthropogenic origin.
How can this be a good indicator of marine air masses?

**Response of the authors:** We have revised the sentences in Line 432-436.

*Comamonas was identified as the dominant bacterium in the coastal atmosphere of*
*Weihai, particularly prevalent in marine air-mass samples. This bacterium is commonly*

*associated with environmental bioremediation and is predominant in oligotrophic environments (Yan et al., 2012; Zhang et al., 2024).*

*Reference*

*Yan, Z., Lingui, X., Lin, L. I.,Hongguang, Z., 2012. Advance in environmental pollutants degradation of Comamonas. Microbiology China. 39, 1471-1478.*

*Zhang, M., B. Zhao, Y. Yan, Z. Cheng, Z. Li, L. Han, Y. Sun, Y. Zheng, and Y. Xia (2024), Comamonas-dominant microbial community in carbon poor aquitard sediments revealed by metagenomic-based growth rate investigation, Science of The Total Environment, 912, 169203, doi:https://doi.org/10.1016/j.scitotenv.2023.169203.*

21. L454-457: "Dust-borne bacteria (Staphylococcus, *Delftia*, *Pseudoalteromonas* and *Deinococcus*) were injected into the atmosphere during dust events, and most of them accompanied the dust transportation to the downwind of Asian Dust including the coastal city of Weihai."

Do the authors have any additional data to support such a statement? Specifically, can you provide $PM_{10}$ concentration data to confirm the occurrence of these dust events?

**Response of the authors:** We have labeled the dust pollution in Figure 8. The MODIS satellite images clearly depicted the transportation of dust particles from the northwest of the study area. This is corroborated by the map of meteorological factors in Figure 9, which suggest a haze-dust synchronous composite pollution. The primary pollutants identified are $PM_{2.5}$ and $PM_{10}$.

**Haze-Dust pollution from March 9 to March 12**

[Figure]

**Figure 8 Satellite images of dust particles air masses transport**

[Figure]

**Figure 9 Air pollutants and meteorological factors of the haze and dust pollution episode**

Pollution episodes occurred in spring from Mar 26 to March 31 was typical dust pollution.

**Dust pollution from March 26 to March 31**

[Figure]

**Figure 10 Satellite images of dust air masses transport (http://worldview.earthdata.nasa.gov).**

[Figure]

**Figure 11 Air pollutants and meteorological factors of the dust pollution**

22. L459-462: "Similarly, microbial communities showed high positively correlated with ions from continental sources, such as $K^+$, $Mg^{2+}$, and $Ca^{2+}$, which indicated that

microorganisms carried by mixed air masses were mostly from continental sources"

The authors claim that microorganisms originate from land because of these correlations. However, they also argue in L298–300 that $Mg^{2+}$ is a typical component of sea salt. While I do not necessarily disagree with their observation, I find it difficult to make such a definitive statement based on their dataset. Did the authors consider performing a simple correlation between the air mass origins and the concentrations of microorganisms?

**Response of the authors:** $K^+$ is the signature ion of biomass combustion (Mason et al., 2016; Yu et al., 2018). $Mg^{2+}$, and $Ca^{2+}$ are mostly derived from crustal elements (Zhang et al., 2012). When the concentrations of the three water-soluble ions, $K^+$, $Mg^{2+}$, and $Ca^{2+}$, are high at the same time, they generally originate from terrestrial sources, including anthropogenic biomass combustion and soil dust.

In addition to anthropogenic sources such as industrial emissions, the sources of $Mg^{2+}$ and $Na^+$ in coastal areas should also take into account the influence of sea salt (Sun et al., 2022). Samples collected under the influence of marine air masses had a moderate correlation between $Mg^{2+}$ and $Na^+$ ($r^2 = 0.67$), indicating that they have a similar origin to sea salt. Moreover, the average $Mg^{2+}/Na^+$ ratio was 0.11, close to the value of 0.12 in seawater (Seinfeld and Pandis, 1997).

We have revised the discussion about $K^+$, $Mg^{2+}$, and $Ca^{2+}$, $Na^+$ in Line 491-498 and Line 508-516.

*Similarly, microbial communities showed high positively correlated with ions from continental sources, such as $K^+$, $Mg^{2+}$, and $Ca^{2+}$ in terrestrial and mixed air mass samples. $K^+$ is the signature ion of biomass combustion (Mason et al., 2016; Yu et al., 2018). $Mg^{2+}$, and $Ca^{2+}$ are mostly derived from crustal elements (Zhang et al., 2012). The dominant bacteria within the microbial community exhibited a significant positive correlation with these three ions simultaneously, suggesting a similar sources. They generally originate from terrestrial sources, including anthropogenic biomass combustion and soil dust.*

*However, a significant positive correlation was observed between Na$^+$ and certain dominant species such as Talaromyces, Monographella, and Phoma (P < 0.05). Similarly, Mg$^{2+}$ was found to be significantly positively correlated with Malassezia in mixed air masses (P<0.01) (Fig. S5). Except for anthropogenic sources, such as industrial emissions, the origins of Mg$^{2+}$ and Na$^+$ in coastal regions should also consider the impact of sea salt (Sun et al., 2022). Samples collected under the influence of marine air masses had a moderate correlation between Mg$^{2+}$ and Na$^+$ (r$^2$ = 0.67), indicating that they have a similar origin of sea salt. Moreover, the average Mg$^{2+}$/Na$^+$ ratio was 0.11, close to the value of 0.12 in seawater (Seinfeld and Pandis, 1997).*

*Reference*

*Mason, P. E., Darvell, L. I., Jones, J. M., and Williams, A.: Observations on the release of gas-phase potassium during the combustion of single particles of biomass, Fuel, 182, 110-117, 10.1016/j.fuel.2016.05.077, 2016.*

*Seinfeld, J. H. and Pandis, S. N.: Atmospheric Chemistry and Physics: From Air Pollution to Climate Change, Wiley: Hoboken, NJ, USA, 1997*

*Sun, H., Sun, J., Zhu, C., Yu, L., Lou, Y., Li, R., and Lin, Z.: Chemical characterizations and sources of PM$_{2.5}$ over the offshore Eastern China sea: Water soluble ions, stable isotopic compositions, and metal elements, Atmos. Pollut. Res., 13, 10.1016/j.apr.2022.101410, 2022.*

*Yu, J., Yan, C., Liu, Y., Li, X., Zhou, T., and Zheng, M.: Potassium: A Tracer for Biomass Burning in Beijing?, Aerosol Air Qual. Res., 18, 2447-2459, 10.4209/aaqr.2017.11.0536, 2018.*

*Zhang, N., Cao, J., Ho, K., and He, Y.: Chemical characterization of aerosol collected at Mt. Yulong in wintertime on the southeastern Tibetan Plateau, Atmos. Res., 107, 76-85, 10.1016/j.atmosres.2011.12.012, 2012.*

23. L462-463: What does a negative correlation with wind direction mean, and how should it be interpreted? I am not sure I understand. Please elaborate.

**Response of the authors:** To discuss the correlation of wind direction and microbial concentration, the wind direction is represented by an angle of 0-360º. The Angle of the wind direction is divided into 360 degrees, the north wind (N) is 0 degrees (i.e. 360

degrees), the east wind (E) is 90 degrees, the south wind (S) is 180 degrees, and the west wind (W) is 270 degrees. Different wind direction means different origins and sources. Therefore, the composition and concentration of bioaerosols was significantly influenced by different source areas (Zhong et al., 2016). This may be important in coastal areas where sea-land breeze can play an important role in microbial community composition and concentration (Huertas et al., 2018).

*Reference*

*Zhong, X., Qi, J., Li, H., Dong, L., and Gao, D.: Seasonal distribution of microbial activity in bioaerosols in the outdoor environment of the Qingdao coastal region, Atmos. Environ., 140, 506-513, 10.1016/j.atmosenv.2016.06.034, 2016.*

*Huertas, M. E., Acevedo-Barrios, R. L., Rodríguez, M., Gaviria, J., Arana, R., and Arciniegas, C.: Identification and Quantification of Bioaerosols in a Tropical Coastal Region: Cartagena de Indias, Colombia, Aerosol Sci. Eng., 2, 206-215, 10.1007/s41810-018-0037-1, 2018.*

24. L464 – 465: "Wind blowing from continent or marine may play important role in microbial community diversity (Jones and Harrison, 2004)"

I am not certain that the wind measured at 15 m above ground level at the sampling station is representative of the entire air mass, but rather reflects a significant local effect. Could the wind have affected the sampling efficiency? How should this effect be interpreted?

**Response of the authors:** Our sampling referred to the requirements of the national standard HJ/T194, and the height of the sampler inlet from the ground was approximately 15 meters during sampling.

According to previous studies, the bacterial ranges were roughly in the same order of magnitude when sampling was affected by wind speed magnitude, and there were significant temporal variations in bacterial concentration and viability (Murata and Zhang, 2016; Sun et al., 2015).

*Reference*

*Murata, K. and Zhang, D.: Concentration of bacterial aerosols in response to synoptic weather and*

*land‑sea breeze at a seaside site downwind of the Asian continent, J. Geophys. Res.: Atmos., 121, 10.1002/2016jd025028, 2016.*

*Sun, Y. L., Wang, Z. F., Du, W., Zhang, Q., Wang, Q. Q., Fu, P. Q., Pan, X. L., Li, J., Jayne, J., and Worsnop, D. R.: Long-term real-time measurements of aerosol particle composition in Beijing, China: seasonal variations, meteorological effects, and source analysis, Atmos. Chem. Phys., 15, 10149-10165, 10.5194/acp-15-10149-2015, 2015.*

25. L468-469: "The marine air masses are generally clean and have a strong scavenging effect on air pollutants."

I disagree, especially in the context of the study where marine air masses are likely transported over nearby lands if integrated over longer periods. The presence of genera such as Streptococcus, Enterobacteriaceae, and Staphylococcus indicates a probable human influence. However, it remains difficult to determine whether this influence is local or if the observed genera are the result of long-distance transport. Additionally, I do not understand this scavenging effect mentioned. At best, there could be a dilution effect, but why would an oceanic air mass remove pollutant? Please rephrase or clarify.

**Response of the authors:** In the coastal city of Weihai, marine air masses predominantly originate from the expansive western Pacific Ocean, located to the east and south. These air masses are typically characterized by their cleanliness and contain a significant quantity of saprophytic and intestinal microorganisms that infiltrate the atmosphere via ocean droplets. Marine air masses from the northerly areas, traverse China's inland Bohai Sea and subsequently pass through urban and rural regions with high human activity levels. Consequently, these areas introduce anthropogenic emission source microorganisms, including pathogenic microorganisms from hospitals and sewage treatment plants.

A correlation analysis between wind speed and $PM_{2.5}$ was conducted. It was also found that $PM_{2.5}$ concentration was negatively correlated with wind speed. high wind speed had a dilution effect on air pollutants.

**Table 1 Correlation analysis table between PM$_{2.5}$ and wind speed**

|     | TOTAL   | TE     | MA    | MIX   |
| --- | ------- | ------ | ----- | ----- |
| r   | -0.731  | -0.770 | -0.049 | -0.738 |
| P   | < 0.01  | 0.009  | 0.039 | 0.037 |

[Figure]

**Figure 12 Scatter plot of wind speed and PM$_{2.5}$**

Pollution elimination was typically initiated with high winds influenced by the cold northwestern airmass form the continent and or from the marine (Fig. 1).

[Figure]

**Fig. 1 Transformation of terrestrial and marine air masses, and chemical composition in PM₂.₅ of three severe air pollution episodes. a, pollution initiation; b, pollution development; c, pollution elimination. TE, terrestrial air mass; MA, marine air mass; MIX, mix air mass.**

26. L474-475: Unless I am mistaken, the temperature was only measured at the sampling site and is not representative of the conditions encountered during transport. Could this relationship indicate a significant site effect?

**Response of the authors:** Thank you for your suggestion. The temperature utilized in this study represents the ambient temperature at the sampling point and does not accurately reflect the temperature of the air mass transport process. The fluctuations in temperature during the air mass transmission process are typically calculated using simulation methods. The sinking process of cold air, a warming process, exhibits dynamic temperature changes that are challenging to measure. In our future research, the meteorological and air quality models will be used to simulate these temperature variations. Therefore, in this study, the ambient temperature near the sampling site was selected.

27. 501-502: "Conversely, the marine air mass facilitates the removal of pollution, introducing a higher number of saprophytic microorganisms"

I am not sure what the authors are suggesting here. Are they implying that airborne microorganisms can remove pollution? Please elaborate. This seems unlikely to me, as several studies show that the contribution of microorganisms to atmospheric chemical processes is very minor compared to physical processes.

**Response of the authors:**
The marine air masses carry low concentrations of air pollutants, while higher wind speeds and higher wind speeds can carry saprophytic bacteria from the sea surface and seawater into marine droplets and then into the atmosphere. The sea breeze can carry these saprophytic bacteria into the atmosphere of coastal cities. At the same time, the sea breeze has a strong dilution effect on the atmospheric particulate matter in coastal

cities, reducing the concentration of atmospheric pollutants in coastal cities (Bagtasa and Yuan, 2020; Pérez et al., 2020).

*Reference*

*Bagtasa, G. and Yuan, C.-S.: Influence of local meteorology on the chemical characteristics of fine particulates in Metropolitan Manila in the Philippines, Atmos. Pollut. Res., 11, 1359-1369, 10.1016/j.apr.2020.05.013, 2020.*

*Pérez, I. A., García, M. Á., Sánchez, M. L., Pardo, N., and Fernández-Duque, B.: Key Points in Air Pollution Meteorology, Int. J. Environ. Res. Public Health, 17, 10.3390/ijerph17228349, 2020.*

Additional minor comments:

28. L39: Please change the too generic word "crucial" or elaborate on why bioaerosols are so crucial.

**Response of the authors:** We have revised the sentence in Line 45-46.

*Bioaerosols, encompassing bacteria, fungi, viruses, pollen, and cellular debris, etc., are vital aerosol particles in the atmosphere.*

29. L124-127: It would be valuable to provide the altitude above sea level as well.

**Response of the authors:** We have added the altitude above sea level in Line 121-122.

*The sampling site was situated at the national air sampling station of Shandong University (37.53°N, 122.06°E), the altitude was 41 m above sea level, and approximately 1-2 km from the coast (Fig. S1).*

30. L245: "The terrestrial air mass accounted for 59.94%" of what? Total air masses reaching the sampling site? Please clarify.

**Response of the authors:** we have revised the sentence in Line 254-256.

*The terrestrial air mass accounted for 59.94% of the total air masses at the sampling site throughout the period, exhibiting an average $PM_{2.5}$ concentration of 36.15±26.52 $\mu g/m^3$.*

31. L448: There is a typo, "concertation was" should read "concentration were"

**Response of the authors:**

We have revised the sentence in Line 478-480.

*Additionally, a positive correlation was observed between bacterial and fungal concentration and $NO_2$ (P<0.05), as well as a significant positive correlation with $PM_{10}$ (P<0.01).*

32. L471: There is a typo. The sentence should read "which suggests an obvious influence from marine sources."

**Response of the authors:** We have revised the sentence in Line 508-509.

*However, a significant positive correlation was observed between $Na^+$ and certain dominant species such as Talaromyces, Monographella, and Phoma (P < 0.05).*

33. L523: There is a typo, a "s" missing to "Staphylococcu"

**Response of the authors:** We have revised the sentence in Line 581-583.

*Bacterial and fungal community influenced by terrestrial and mixed air masses were enriched in animal pathogens, including the Staphylococcus, Malassezia and Alternaria.*

34. Figure 1: Why is there two separate stacked histograms for each event? What does the legend of the graph represent? I'm guessing the altitude, but then why is it negative (-274)?

**Response of the authors:** The two separate stacked histograms represented the concentrations of water-soluble ions and metal elements, which are split into the left (water-soluble ions) and right (metal elements).

We have checked the legend of the map in the software of ArcGIS and revised the scale range based on the actual altitude in Fig. 1.

[Figure]

**Fig. 1 Transformation of terrestrial and marine air masses, and chemical composition in PM$_{2.5}$ of three severe air pollution episodes. a, pollution initiation; b, pollution development; c, pollution elimination. TE, terrestrial air mass; MA, marine air mass; MIX, mix air mass.**

35. Figure 2: The legend should clearly specify the criteria used for selecting the main genera.

**Response of the authors:** We have modified the legend in Figure 2. The genera in Fig. 2 were the top 15 and top 10 species for bacteria and fungi based on the relative abundance.

36. Figure 3: Would it possible to display clearly the number of each air masses (n) computed in the statistical analyze? What about the error bars?

**Response of the authors:** We have modified Fig.3 and added the error bars. The information of samples is summarized in Table S1, terrestrial air mass samples (10), marine air mass samples (6), and mixed mass samples (8). In the Wilcoxon rank-sum test for two groups, the difference between proportions is denoted in the figure, indicating the range of variation in its relative abundance.

[Figure]

**Fig.3 Bacterial (a) and fungal (b) community disparities influenced by terrestrial (n=10), marine (n=6), and mixed mass (n=8).**

37. Figure 4: A P value of 0.0522 was displayed on Figure a. Since it's above the significant threshold of 0.05 set up in the statistical part, I suggest to use a different nomenclature if the authors still want to point out that the difference is nearly significant. Especially considering that test can't tell which of the air masses are different from the other. Also what about the error bars and number of individuals (n)?

**Response of the authors:** Thank you for your suggestions. We have modified Fig.3 and added the error bars. The information of samples is summarized in Table S1, terrestrial air mass samples (10), marine air mass samples (6), and mixed mass samples (8). In the Kruskal-Wallis H test, *$P$< 0.05, **$P$<0.01 indicate significant differences. Meanwhile, the horizontal coordinates represent the relative abundance of species or ecological functions, and comparisons between multiple groups often show that the

results are not significantly different overall (*P*>0.05), but their relative abundance varies greatly. In two groups comparison, the results may be significant (e.g., *P* = 0.1498 for *Aspergillus* when comparing between three groups, while the comparison of fungi under the influence of marine and mixed air masses yielded a result of *P* = 0.00574).

[Figure]

**Fig. 4 Bacterial (a) and fungal (b) community function disparities influenced by terrestrial (n=10), marine (n=6), and mixed mass (n=8).**

38. Figure 5: The legend shows a Pearson r coefficient, but it is stated in the statistical analysis section that Mantel tests were computed using Spearman's rank correlation. Please rectify. Also, please clarify what "Cells" refer to, I didn't find it anywhere in the

manuscript.

**Response of the authors:** We have modified Figure 5. The correlation coefficient in Materials and Methods should be "Pearson's r", which we have corrected in the manuscript. "Cells" refers to the abundance of bacterial and fungal concentration based on qPCR.

*The Mantel analysis was utilized to reveal the correlation between microbial community and various environmental factors. The Pearson's r coefficient at p<0.05 and p<0.01 indicates the significant correlation.*

[Figure]

**Fig. 5 Mantel correlation reveal the relationships between microorganisms and environmental factors influenced by different air masses, terrestrial (a), marine (b) and mixed air masses (c).**

**Response to reviewer 2**

The manuscript submitted by He et al. has a considerable and original work to clarify the interactions between bioaerosols, airborne chemicals and their sources. Although I would consider it eligible to publish, some major explanations and analyses are needed, with special attention to writing and organization of the manuscript, and some disagreement that should be discussed or clarified.

**Response of the authors:** We greatly appreciate the positive comments and constructive suggestions on our manuscript. According to your comments and suggestions, we have made detailed revision and incorporated them into the revised manuscript.

Major changes

1. - Introduction section. Lines 83-116 provides very specific and detailed information, which corresponds (and can be used) for a formal discussion section. I suggest to shorten this part, keeping the main ideas to justify the work and set up the objectives but with less details of previous works.

**Response of the authors:** Thank you for your suggestion. We have condensed this section and addressed within the discussion section.

2. - Materials and Methods. A representative reference for chemical elements analyses would be appreciated, since no much details are provided. FAPROTAX and FUNGuild references should be specified.

**Response of the authors:** The reference and detailed information for chemical elements analyses, FAPROTAX and FUNGuild, have been added in the revised manuscript in Line 149-164 and Line 227-234.

*The mass concentration of PM$_{2.5}$, water-soluble ions, and metal elements were quantified after sampling. The membranes were meticulously weighed utilizing a Mettler XP-6 balance with an accuracy of 10$^{-6}$ g. Prior to the weighing, the membranes were maintained in a controlled environment with consistent temperature and humidity*

for a duration of 24 hours. Ion chromatography (ICS-2100, Chameleon 6. 8, AS-DV autosampler Thermo Fisher) was employed to determine the concentration of water-soluble ions such as $Na^+$, $K^+$, $Ca^{2+}$, $Mg^{2+}$, $Cl^-$, $NO_3^-$, $SO_4^{2-}$, and $NH_4^+$. These ions were extracted by sonication with ionized water for 45 min and then separated by anion (IonPacAS23) or cation column exchange (IonPacCS12A). Then, these ions were detected using a conductivity detector with an anion separation column of IonPacAS23, flow rate: 1.0 mL/min, an anion suppressor of AERS500, and conductivity detector. The injection volume was 25 μL, and the cation separation column was IonPacCS12A, flow rate:1.0 mL/min, cation suppressor was CERS500 (Zhang et al., 2022). Metallic elementals including Al, Fe, Ti, Mn, Co, Ni, Cu, Zn, Ga, Sr, Cd, Sn, Sb, Pb, V, Cr, and As, were extracted using microwave digestion extraction (ETHOS ONE, Milestone), with the concentrations determined by ICP-MS or ICP-OES (Thermo Fisher).

Bacterial community functional was conducted using FAPROTAX, a manually constructed database that maps prokaryotic taxa to metabolic or other ecologically functions, such as sulfur, nitrogen, hydrogen, and carbon cycling (Chen et al., 2022). FUNGuild (Fungi Functional Guild) was used to predict the fungal ecological function. This tool could classify and analyze fungal communities by the microecological guild based on current published literature or data from authoritative websites to classify fungi functionally (Nguyen et al., 2016). Three primary groups are obtained based on the nutritional mode: Pathorotroph, Symbiotroph, and Saprotroph.

Reference

Nguyen, N. H., Song, Z., Bates, S. T., Branco, S., Tedersoo, L., Menke, J., Schilling, J. S., and Kennedy, P. G.: FUNGuild: An open annotation tool for parsing fungal community datasets by ecological guild, Fungal Ecol., 20, 241-248, 10.1016/j.funeco.2015.06.006, 2016.

Chen, J., Zang, Y., Yang, Z., Qu, T., Sun, T., Liang, S., Zhu, M., Wang, Y., and Tang, X.: Composition and Functional Diversity of Epiphytic Bacterial and Fungal Communities on Marine Macrophytes in an Intertidal Zone, Front. Microbiol., 13, 10.3389/fmicb.2022.839465, 2022.

*Zhang, Y., Guo, C., Ma, K., Tang, A., Goulding, K., and Liu, X.: Characteristics of airborne bacterial communities across different PM$_{2.5}$ levels in Beijing during winter and spring, Atmos. Res., 273, 10.1016/j.atmosres.2022.106179, 2022.*

- Results and Discussion.

3. I think the manuscript would benefit from an appropriate Discussion section, separated from Results. Comments from some results are too long and descriptive to follow the results fluently, and also some marks are repetitive, as Deinococcus (lines 348 and 388), Comamonas (lines 367 and 394). I would suggest split Results from Discussion, making the former shorter and focused on the data, and use the information to elaborate a discussion properly, without duplicities and allowing you to explain the relationships and hypotheses in a more consistent fashion.

**Response of the authors:** Thank you for your suggestion. The separated "Results" and "Discussion" section was in the revised manuscript.

4. The title "3.1 Air mass backward trajectory" does not correspond properly with the content described in this section.

**Response of the authors:** Thank you for your suggestion. We have revised the title.

*3.1 Air masses categorization and typical pollution processes*

5. Several times, "spring season" (L262, L309, L474, L491, L536) is cited in the manuscript, but the sampling was conducted mostly in winter days. It is confusing. These references are based on other works? Did one of the Pollution episodes occur in Spring? Explain it.

**Response of the authors:** Winter and spring in northern China are characterized by a high incidence of haze and dust pollution. During the winter months, specifically from January to February, temperatures are low. This period is the central heating period of northern China, coal burning and other heating methods will lead to the increase of air pollutant emission intensity, and haze pollution incident is more serious. As spring

progresses into March, temperatures and wind speed begin to rise. Northern China enters the spring dust season. Notably, the escalating frequency of extreme dust events in recent years has facilitated the transport of regional dust aerosols, significantly impacting both the northern and southern regions of China. In this study, pollution episodes occurred in spring from Mar 9 to March 12 was typical Haze‑Dust composite pollution. Pollution episodes occurred in spring from Mar 26 to March 31 was typical dust pollution.

**Haze-Dust composite pollution from March 26 to March 31**

[Figure]

Figure 1 Satellite images of haze and dust air mass transport from March 9 to March 12

[Figure]

Figure 2 Air pollutants and meteorological factors of the haze-dust composite pollution

**Dust pollution from March 26 to March 31**

[Figure]

Figure 3 Satellite images of dust air masses transport (http://worldview.earthdata.nasa.gov).

[Figure]

**Figure 4 Air pollutants and meteorological factors of the dust pollution**

6. Fig. 1. Please indicate the date of the events and the different scale for left and right axes.

**Response of the authors:** We have indicated the date of the events in Fig 1. The different scale for left and right axes are indicated in the figure legend. The concentration of water-soluble ions is indicated in the left axes. The concentration of metal elements (Cu, Zn, Al, Fe) is indicated in the right axes.

[Figure]

**Fig. 1 Transformation of terrestrial and marine air masses, and chemical composition in PM$_{2.5}$ of three severe air pollution episodes. a, pollution initiation; b, pollution development; c, pollution elimination. TE, terrestrial air mass; MA, marine air mass; MIX, mix air mass.**

7. NMDS or PCoA, and their respective ANOSIM or PERMANOVA analyses would increase the support some conclusions.

**Response of the authors:** We have added the Principal Coordinates Analysis (PCoA) in Fig.S4. Here, PCoA was conducted to examine the overarching differences among samples influenced by different air masses (Figure 10). PCoA is an unconstrained method of dimensionality reduction analysis that can be employed to explore similarities or disparities in the composition of microbial communities. As depicted in Figure 12, the $PM_{2.5}$ samples from each group are closely clustered, suggesting that the bacterial community structure of $PM_{2.5}$ remains consistent across groups. Conversely, the $PM_{2.5}$ samples influenced by terrestrial air masses are more dispersed, with greater distances between samples. This finding, to a certain degree, indicates variations in the airborne bacterial communities under different air mass influences.

[Figure]

**Fig.S4 PCoA analysis of bacterial and fungal community under different air masses**

**terrestrial air masses (blue), marine air masses (green), and mixed air masses(red)**

8.Fig. 2. I am guessing that left axis of the graphs are not on 100% but 1%. How the authors explain the substantial change in fungal communities? For instance, Aspergillus completely disappeared from the top abundance in MIX when it was the top one for TE and MA.

**Response of the authors:** We have modified Fig. 2. The X-axis represents the percentage of relative abundance, which adds up to 100%.

[Figure]

**Fig. 2. Bacterial and fungal species and function influenced by different air masses. Bacterial and fungal community concentration, main phylum (a), (b), genus (c), (d), and community function (e), (f) are indicated.**

*Aspergillus* generally has a higher abundance in cleaner samples. In the figure, it shows the highest relative abundance in marine air mass samples, followed by terrestrial air masses, and the lowest relative abundance in mixed air masses. *Aspergillus* was 53.7% and 20.1% in marine and terrestrial air-mass samples, respectively. *Aspergillus* is a dominant fungus in offshore areas such as Qingdao, China (Li et al., 2011). Moreover, the Saprophytic *Aspergillus* was also prevalent in clean samples during haze pollution episode and was commonly detected on non-Haze days (Yan et al., 2016). Prior research has established that *Aspergillus* is ubiquitously found in nature and non-polluted

environments (Li and Kendrick, 1995).

For the coastal city of Weihai, we examined the relative abundance of *Aspergillus* in terrestrial and mixed air mass samples and found that the abundance was low in contaminated samples from terrestrial and contaminated samples from mixed samples (Figure 5). Moreover, we compared the abundance of *Aspergillus* in samples collected from coastal and inland cities within Shandong Province during the same sampling period, and found that in the inland city of Jinan (a highly polluted inland city), the abundance of *Aspergillus* was much higher than in the coastal city of Weihai. This indicates that *Aspergillus* has a relatively high abundance in low pollution ambient air.

[Figure]

**Figure 5 The top 15 abundant fungi in PM$_{2.5}$ at genus level in inland and coastal cities.**

*Reference*

*Li, D.-W. and Kendrick, B.: A year-round study on functional relationships of airborne fungi with meteorological factors, Int. J. Biometeorol., 39, 1995.*

*Li, M., Qi, J., Zhang, H., Huang, S., Li, L., and Gao, D.: Concentration and size distribution of bioaerosols in an outdoor environment in the Qingdao coastal region, Sci. Total Environ., 409, 3812-3819, 10.1016/j.scitotenv.2011.06.001, 2011.*

*Yan, D., Zhang, T., Su, J., Zhao, L.-L., Wang, H., Fang, X.-M., Zhang, Y.-Q., Liu, H.-Y., and Yu, L.-Y.: Diversity and Composition of Airborne Fungal Community Associated with Particulate Matters in Beijing during Haze and Non-haze Days, Front. Microbiol., 7, 10.3389/fmicb.2016.00487, 2016.*

9. Lines 425-428 are hard to interpretate. Please revise. Are they a conclusion or supported information to your results?

**Response of the authors:** This section is the conclusion of the FAPROTAX analysis. Overall, the dominant airborne bacteria in the coastal city primarily inhabited anthropogenic environments such as soil, water, and terrestrial ecosystems. Additionally, marine ecosystems served as a significant source of airborne microbes.

10. Section 3.4 Community disparities influenced by terrestrial, marine and mixed air masses. The disparities mentioned in L387-388 are not supported statistically according to Fig. 3 or the p-values numbers are wrong. The same goes for the fungal graphs and text, for which the p-values do not concord (*Aspergillus*, p=0.014 or p=0.1498?; *Malassezia*, p=0.041 or 0.047??).

**Response of the authors:** We rechecked Fig. 3 and revised the section in Line 442-453. In the comparison of three types of air mass samples, the p value of *Aspergillus* was p=0.1498, *Malassezia* was p= 0.047. However, when comparing the oceanic and mixed air masses, the p Value of *Aspergillus* was p=0.00574, which showed significantly different between marine and mixed air masses. Comparisons between multiple groups often result in overall non-significant differences in the results (p>0.05), but with large variations in the relative abundance. In subsequent two-by-two comparisons, the results may be more significant (p<0.05). This difference maybe mainly caused by the different statistical analysis strategies used in multi group comparisons and two group comparisons.

*For fungal community, Aspergillus demonstrated a significant differentiation between marine and mixed airmasses (P=0.005). The highest proportion was noted in samples from marine air masses, at 53.7%. In contrast, the values were 20.1% and 0.3% in terrestrial and mixed air masses respectively. Previous studies have confirmed that Aspergillus is widely distributed in nature and unpolluted environments (Kendrick, 1995). This fungus is predominantly found in offshore regions, such as Qingdao, China (Li et al., 2011). Furthermore, the saprophytic Aspergillus was also prevalent in clean*

*samples during periods of haze pollution and was frequently detected on non-haze days (Yan et al., 2016). Malassezia was higher in terrestrial and mixed air-mass samples (P=0.047), which has been found to be widespread in a variety of animals. As a parasitic fungus, Malassezia causes the majority of skin diseases, such as dandruff and seborrheic dermatitis caused by Malassezia sphericalis (Deangelis et al., 2007).*

11. Fig. 5 shows correlations with "Bacteria", "Fungi", and "Cells". Firstly, the legend shows "Pearson's r", when in Materials and Methods is specified Spearman's. Please clarify this point. Bacteria and Fungi correlations with chemical elements are referred to concentrations or community compositions? "Cells" are the direct sum of bacterial and fungal concentrations?

**Response of the authors:** The "Pearson's r" was used in the mantel analysis. We have corrected the expression in the Materials and Methods section in Line 246-248.

The correlation of bacteria and fungi with chemical elements refers to community composition, and the top 15 genera were screened for correlation analysis with chemical elements and Meteorological conditions. The "cells" is the abundance of bacterial and fungal concentrations based on qPCR. We have modified the expression with "microbial concentration".

*The Mantel analysis was utilized to reveal the correlation between microbial community and various environmental factors. The Pearson's r coefficient at p<0.05 and p<0.01 indicates the significant correlation.*

[Figure]

**Fig. 5 Mantel analysis reveal the correlation between microbial community and various environmental factors under different air masses, terrestrial air masses (a) marine air masses (b) and mixed air masses (c). The Pearson's r correlation coefficient indicates the significant correlation at p<0.05 and p<0.01.**

12. $PM_{2.5}$ and $PM_{10}$ concentrations were positive correlated. However, the authors found a negative correlation with $PM_{10}$ concentrations but not with $PM_{2.5}$. Moreover, several studies have shown a positive correlation between richness and microbial concentrations with PMs concentrations. The authors should elaborate a discussion about this. Are the correlations maintained by group of microorganisms (bacteria or fungi) or this is only observed when the addition of both is conducted?

**Response of the authors:** The positive correlation with $PM_{10}$ is mainly due to the contribution of surface coarse particles to the atmospheric microbial community during

sand dust events in spring. Under the influence of strong winds, the surface soil is the main source of coarse particles ($PM_{10}$). Thus, a positive correlation between airborne microorganisms and $PM_{10}$. During the dust period, the concentration of fine particles ($PM_{2.5}$) was lower compared to $PM_{10}$, thus showing a non-significant correlation between microbial concentration and $PM_{2.5}$.

Pollution episodes occurred in spring from Mar 26 to March 31 was typical dust event. The MODIS true-color images revealed a transit of highly polluted air masses originating from Northwest of the coastal city from March 26. Dust particles were then carried downwind to the eastern coastal city of Weihai on March 29 and 30. A significant increase in $PM_{10}$ concentration was observed, with an hourly maximum value of 197 μg/m$^3$. The relatively low $PM_{2.5}/PM_{10}$ ratio of 0.28 indicated pronounced dust pollution. Historical radar map analysis revealed that the primary pollutants was $PM_{10}$ on March 30.

**Dust pollution from March 26 to March 31**

[Figure]

**Figure 6 Satellite images of dust air masses transport (http://worldview.earthdata.nasa.gov).**

[Figure]

**Figure 7 Air pollutants and meteorological factors of the dust pollution**

13. RDA analysis with the environmental and physicochemical parameters would be complementary to correlations.

**Response of the authors:** We have added the RDA analysis in the supplementary materials Fig. S4. Bacterial community composition under the influence of marine air masses were positively correlated with humidity while bacteria under the influence of terrestrial air masses were positively correlated with $PM_{2.5}$ and $PM_{10}$, and other crustal elements, such as $Mg^{2+}$, $Ga^{2+}$, $K^+$.

[Figure]

**Fig. S4 RDA of bacterial and fungal community structure with environmental parameters.**

14. L469: "The marine air masses are generally clean and have a strong scavenging effect on air pollutants". From Fig 5., Wind speed is positively correlated with PMs

concentrations in marine air masses (higher speed higher concentrations) and negatively with Continental ones. How does it fit with the clearance effect the authors propose?

**Response of the authors:** The sampling period predominantly occurs during the winter and early spring seasons in northern China, characterized by a systematic northwest wind direction. Therefore, the air masses frequently originate from terrestrial and mixed sources. The statistical examination of terrestrial and mixed air masses reveals a negative correlation between $PM_{2.5}$ concentration and wind speed, suggesting the strong scavenging effect of high winds on $PM_{2.5}$. During the study period, there were a limited number of typical marine air masses, with only 6 samples being selected for analysis. By further screening the samples, the number of samples affected by ocean air masses was increased, and a correlation analysis between wind speed and $PM_{2.5}$ was conducted. It was also found that $PM_{2.5}$ concentration was negatively correlated with wind speed.

Table 1 Correlation analysis table between $PM_{2.5}$ and wind speed

|   | TOTAL | TE | MA | MIX |
| --- | --- | --- | --- | --- |
| r | -0.731 | -0.770 | -0.049 | -0.738 |
| P | < 0.01 | 0.009 | 0.039 | 0.037 |

[Figure]

**Figure 8 Scatter plot of wind speed and $PM_{2.5}$**

15. Lines 480-509 reminds a mixture of Discussion and Conclusions. This would be solved by separating Results and Discussion.

**Response of the authors:** The separated Results and Discussion sections are provided in the revised manuscript.

16.Minor changes

Typos:

L324: a point (.) before reference.

**Response of the authors:** We have revised the sentence in Line 333-336.

*The V/Ni ratio is employed as a measure of the influence of ship emissions. A ratio exceeding 0.7 typically indicates a significant impact from these emission sources, and is commonly used as an indicator in coastal cities (Zhang et al., 2014).*

L342 Actionbacteria--> Actinobacteria

**Response of the authors:** We have revised the sentence in Line 349-351.

*Predominantly, Proteobacteria (40.06%), Firmicutes (36.30%), Actinobacteria (8.97%), Bacteroidota (8.29%), and Deinococcus-Thermus (4.59%) were identified as the most abundant bacteria.*

L448 Concertation--> Concentration

**Response of the authors:** We have revised the sentence in Line 478-480.

*Additionally, a positive correlation was observed between bacterial and fungal concentrations and $NO_2$ (P<0.05), as well as a significant positive correlation with $PM_{10}$ (P<0.01).*

---

## Referee Report (RR1)

**Reviewer Comments on Authors' Revisions**

I would like to thank the authors of the manuscript entitled "*Influence of terrestrial and marine air-mass on the constituents and intermixing of bioaerosols over coastal atmosphere*" for taking the time to respond clearly and in detail to all the comments. I find that the manuscript has been significantly improved and, I believe, deserves to be published in Atmospheric Chemistry and Physics. That being said, I have additional comments that I would appreciate you considering.

L338-347 and answer to the comment 15 of the authors:

I am not criticizing the qPCR detection method at all, as it is a widely validated method in the literature. I am simply questioning the semantics used: qPCR does not provide information on the number of cells per m³ of air, but rather on the number of 16S genes, usually expressed as copies per m³ of air. Most bacteria have multiple copies of the 16S gene in their genome, ranging from 1 to 15. In your case, the methodology used (employing *E. coli* as a standard) allows you to approximate a cellular concentration, but it remains an estimation. I would rather refer to genome copies per m³ of air.

There are also many spelling errors and typos in this manuscript. I suggest a thorough review to correct them all. Here are the ones I noted:

L60: there is a comma missing after "*environment*".

L71: "*maybe*" should read "*may be*".

L74: seems there is a word missing after "*marine*". Maybe the authors mean "*marine environment* or *surface*"?

L104: "*costal*" should read "*coastal*".

L124: "*between January to March*" should read "*between January and March*" or "*from January to March*".

L127: there is a comma missing after "*utilized*".

L128: "*ion*" should read "*ions*".

L132: the tilde should be replaced by "*to*" (*7:00 to 18:30*).

L161-162: "*metallic elementals*" should read "*metallic elements*".

L169: "*were simulated one-hour intervals*" should read "*were simulated at one-hour intervals*".

L176: "*and mixed air-mass sample*" should read "*and a mixed air-mass sample*" .

L227: "*Bacterial community functional was conducted*" should read "*Bacterial community functional analysis was conducted*".

L228-229: "*other ecologically functions*" should read "*other ecological functions*".

L234: « *Pathorotroph* » should read « *Pathotroph*".

L235: "*different airmasses*" should read "*different air masses*".

L247-248: "*microbial community and various environmental factors*" should read "*microbial community composition and various environmental factors*".

L257: "*During reginal haze pollution*" should read "*During regional haze pollution*".

L281: "*Cold northwestern airmass form the continent*" should read "*Cold northwestern airmass from the continent*".

L288: "*northwestern airmass form the continent*" should read "*northwestern airmass from the continent*".

L302: please replace "*succeeded*" by "*followed*".

L309: "*with a range that from*" should read "*that ranged from*" or "*with a range of*".

L310: "*ion*" should read "*ions*".

L315: "*The concentration of $K^+$ 0.24±0.20 μg/m³*" should read "*The concentration of $K^+$ was 0.24±0.20 μg/m³*".

L324: "*coal combination in winter heating*" probably did the authors mean "*coal combustion*"?

L356: "*in both terrestrial and mixed air-mass samples*" should read "*in both terrestrial and mixed air-masses samples*".

L366: please replace "*including*" by "*included*".

L372: "*to the previously studies*" should read "*to the previous studies*".

L388: I guess that "*and automatic compound*" should read "*and aromatic compound*".

L394: "*the prevalence of Saprotroph fungi was observed higher in samples*" should read "*the observed prevalence of Saprotroph fungi was higher in samples*".

L406: there is one extra parenthesis.

L408: "*airmasses*" should read "*air masses*".

L413: please replace "*conducted*" by "*shown*".

L415: "air mass" should read "air mass samples".

L467: "*automatic compound degradation bacteria*" should read "*aromatic compound degradation bacteria*".

L482: "*diverse and abundance microbial populations*" should read "*diverse and abundant microbial populations*".

L488-490: "*...accompanied the dust transportation to the downwind of Asian Dust including the coastal city of Weihai. Influenced by mixed air masses, bacterial community was 490 significantly positively correlated with K+ (P<0.01) and PM10 (P<0.05)*".

The sentence is weird.

L491: "*showed high positively correlated*" should read "*showed high positive correlation*" or "*were positively correlated*".

L511: Replace "*Except*" » by "*Besides*"?

L517: "*temperature have*" should read "*temperature has*".

L519: "*marine airmass samples*" should read "*marine air mass samples*".

---

## Author Response (AR2)

**Influence of terrestrial and marine air-mass on the constituents and intermixing of bioaerosols over coastal atmosphere**

Qun He[a], Zhaowen Wang[a], Houfeng Liu[a], Pengju Xu[a], Rongbao Duan [a], Caihong Xu[b],

Jianmin Chen[b], Min Wei[a,b,*]

[a] College of Geography and Environment, Shandong Normal University, Ji'nan 250014, China

[b] Shanghai Key Laboratory of Atmospheric Particle Pollution and Prevention (LAP³), Fudan

Tyndall Centre, Department of Environmental Science & Engineering, Fudan University,

Shanghai 200433, China

* Correspondence to: Min Wei (minwei@sdnu.edu.cn)

**Response to editor**

We thank the editor for the opportunity to revise our manuscript, we appreciate the editor and reviewer very much for their positive and constructive comments and suggestions. We have revised our manuscript according to the reviewer's comments, and the response to the reviewer's comments are in detailed below. We also attach a revised manuscript with tracked changes, and the amendments were marked in red in the revised manuscript.

**Response to reviewer**

I would like to thank the authors of the manuscript entitled "Influence of terrestrial and marine air-mass on the constituents and intermixing of bioaerosols over coastal atmosphere" for taking the time to respond clearly and in detail to all the comments. I find that the manuscript has been significantly improved and, I believe, deserves to be published in Atmospheric Chemistry and Physics. That being said, I have additional comments that I would appreciate you considering.

**Response of the authors:**

We greatly appreciate the positive comments and constructive suggestions on our

manuscript. According to your suggestions, we have carefully incorporated them into our paper and made detailed revision based on your comments.

1. L338-347 and answer to the comment 15 of the authors: I am not criticizing the qPCR detection method at all, as it is a widely validated method in the literature. I am simply questioning the semantics used: qPCR does not provide information on the number of cells per m³ of air, but rather on the number of 16S genes, usually expressed as copies per m³ of air. Most bacteria have multiple copies of the 16S gene in their genome, ranging from 1 to 15. In your case, the methodology used (employing E. coli as a standard) allows you to approximate a cellular concentration, but it remains an estimation. I would rather refer to genome copies per m³ of air.

**Response of the authors:**

We thank the reviewer for the comments and suggestions. QPCR does not quantify the number of cells per m³ of air, but rather the number of 16S genes, usually expressed as copies per m³ of air. Most bacteria have multiple copies of the 16S gene in their genome. We used the data on bacterial rRNA gene copy number from the rrnDB database (https://rrndb.umms.med.umich.edu/) (Stoddard et al., 2015). According to the latest data, the copy number of the 16S rRNA gene in their genome ranges from 1 to 21 with an average value of 5.5.

The number of bacterial cells, calculated based on the mean copy number of 5.5 in the bacterial genome, is an estimate value. The rRNA copies per genome may range from 50 to 100 in filamentous fungi (Rooney and Ward, 2005) and ascomycetes often have relatively smaller genome sizes (Kullman et al., 2005). The copy number of the fungal ITS gene in their genome was 50 in this study.

As suggested by the reviewers, we have changed the unit of bacterial and fungal number to copies per m³ of air. The Figure 2, and Section 3.3 Bacterial and fungal count, have been revised according to the unit of copies per m³ of air.

[Figure]

Figure 1 The estimate of copy number of 16S rRNA genes in domain Bacteria

**Reference**

*Kullman, B., Tamm, H., and Kullman, K.: Fungal genome size database, http://www.zbi.ee/fungal-genomesize, 2005.*

*Rooney, A. P. and Ward, T. J.: Evolution of a large ribosomal RNA multigene family in filamentous fungi: birth and death of a concerted evolution paradigm, Proc. Natl. Acad. Sci. USA, 102, 5084-5089, https://doi.org/10.1073/pnas.0409689102, 2005.*

*Stoddard, S. F., Smith, B. J., Hein, R., Roller, B. R. K., and Schmidt, T. M.: rrnDB: improved tools for interpreting rRNA gene abundance in bacteria and archaea and a new foundation for future development, Nucleic Acids Res., 43, D593-D598, https://doi.org/10.1093/nar/gku1201, 2015.*

There are also many spelling errors and typos in this manuscript. I suggest a thorough review to correct them all. Here are the ones I noted:

**Response of the authors:** Thank you for highlighting the spelling errors in our manuscript. We have reviewed and revised the manuscript, and the spelling errors have been corrected.

2. L60: there is a comma missing after "environment".

**Response of the authors:** We have revised the sentence in Line 59-61.

*The geographical and topographical factors, such as terrestrial and marine environments, exhibit significant differences in bioaerosol sources and pollution characteristics.*

3. L71: "maybe" should read "may be".

**Response of the authors:** We have revised the sentence in Line 71-73.

*Bioaerosols from the oceans may be influenced by long-distance transport from continental sources, such as plants and human pathogens (Elbert et al., 2007; Sharoni et al., 2015).*

4. L74: seems there is a word missing after "marine". Maybe the authors mean "marine

environment or surface"?

**Response of the authors:** We have revised the sentence in Line 73-76.

*Studies have shown that the concentration and diversity of bacterial and fungal aerosols from marine environment are typically lower than those derived from continental sources (Cao et al., 2024; Xue et al., 2022; Shi et al., 2022).*

5. L104: "costal" should read "coastal".

**Response of the authors:** We have revised the sentence in Line 104-106.

*The coastal aerosols provide the ideal conditions for understanding the mixing processes taking place between natural and anthropogenic air masses from terrestrial and marine.*

6. L124: "between January to March" should read "between January and March" or "from January to March".

**Response of the authors:** We have revised the sentence in Line 124-125.

*$PM_{2.5}$ samples were gathered between January and March, 2018, during the winter heating and spring dust seasons in northern China.*

7. L127: there is a comma missing after "utilized".

**Response of the authors:** We have added the comma in Line 125-129.

*Two parallel $PM_{2.5}$ samplers (TH-150C-III, Wuhan Tianhong Instrument Co., Ltd., China) with a particle size of 2.5 ± 0.2 μm were utilized, equipped with Quartz membrane for the collection of $PM_{2.5}$, inorganic ions, metal elements, and microorganisms at a flow rate of 100 L $min^{-1}$.*

8. L128: "ion" should read "ions".

**Response of the authors:** We have revised the sentence in Line 125-129.

*Two parallel $PM_{2.5}$ samplers (TH-150C-III, Wuhan Tianhong Instrument Co., Ltd., China) with a particle size of 2.5 ± 0.2 μm were utilized, equipped with Quartz membrane for the collection of $PM_{2.5}$, inorganic ions, metal elements, and microorganisms at a flow rate of 100 L $min^{-1}$.*

9. L132: the tilde should be replaced by "to" (7:00 to 18:30).

**Response of the authors:** We have revised the sentence in Line 129-132.

*The Quartz membranes were cauterized in a muffle furnace at 450°C for 6 hours to remove carbonaceous and contaminant materials. Intermittent sampling was used and samples were collected twice a day (7:00 to 18:30 and 19:00 to 06:30 the following day).*

10. L161-162: "metallic elementals" should read "metallic elements".

**Response of the authors:** We have revised the sentence in Line 161-164.

*Metallic elements including Al, Fe, Ti, Mn, Co, Ni, Cu, Zn, Ga, Sr, Cd, Sn, Sb, Pb, V, Cr, and As, were extracted using microwave digestion extraction (ETHOS ONE, Milestone), with the concentrations determined by ICP-MS or ICP-OES (Thermo Fisher).*

11. L169: "were simulated one-hour intervals" should read "were simulated at one-hour intervals".

**Response of the authors:** We have revised the sentence in Line 169-170.

*Backward trajectories were simulated at one-hour intervals and estimated over a 24-hour period.*

12. L176: "and mixed air-mass sample" should read "and a mixed air-mass sample".

**Response of the authors:** We have revised the sentence in Line 173-178.

*In this study, air mass categories were defined that if more than 90% of the masses originated from the ocean it was a typical sample affected by marine air-mass; a typical terrestrial air-mass sample if more than 90% originated from the continent; and a mixed air-mass sample if the proportions of terrestrial and marine air masses were similar or accounted for more than 40% of the total air masses in one day.*

13. L227: "Bacterial community functional was conducted" should read "Bacterial community functional analysis was conducted".

**Response of the authors:** We have revised the sentence in Line 235-238.

*Bacterial community functional analysis was conducted using FAPROTAX, a manually constructed database that maps prokaryotic taxa to metabolic or other ecological functions, such as sulfur, nitrogen, hydrogen, and carbon cycling (Chen et al., 2022).*

14. L228-229: "other ecologically functions" should read "other ecological functions".

**Response of the authors:** We have revised the sentence in Line 235-238.

*Bacterial community functional analysis was conducted using FAPROTAX, a manually constructed database that maps prokaryotic taxa to metabolic or other ecological functions, such as sulfur, nitrogen, hydrogen, and carbon cycling (Chen et al., 2022).*

15. L234: « Pathorotroph » should read « Pathotroph".

**Response of the authors:** We have revised the sentence in Line 241-242.

*Three primary groups are obtained based on the nutritional mode: Pathotroph, Symbiotroph, and Saprotroph.*

16. L235: "different airmasses" should read "different air masses".

**Response of the authors:** We have revised the sentence in Line 243-244.

*Samples affected by different air masses were examined for intergroup species variability, based on community abundance data.*

17. L247-248: "microbial community and various environmental factors" should read "microbial community composition and various environmental factors".

**Response of the authors:** We have revised the sentence in Line 254-255.

*The Mantel analysis was utilized to reveal the correlation between microbial community composition and various environmental factors.*

18. L257: "During reginal haze pollution" should read "During regional haze pollution".

**Response of the authors:** We have revised the sentence in Line 265-267.

*During regional haze pollution, the terrestrial air masses primarily influenced Weihai were typically originating from the Beijing-Tianjin-Hebei region and the surrounding areas.*

19. L281: "Cold northwestern airmass form the continent" should read "Cold northwestern airmass from the continent".

**Response of the authors:** We have revised the sentence in Line 289-290.

*Cold northwestern airmass from the continent, and marine air masses from east or south, were the primary contributors during pollution mitigation.*

20. L288: "northwestern airmass form the continent" should read "northwestern airmass from the continent".

**Response of the authors:** We have revised the sentence in Line 295-297.

*Pollution elimination was initiated with high winds, snowfall influenced by the cold northwestern airmass from the continent and marine air masses from the northeast sea.*

21. L302: please replace "succeeded" by "followed".

**Response of the authors:** We have revised the sentence in Line 307-312.

*In marine air-mass samples, a lower concentration of water-soluble ions was observed, with the concentration of 13.01±7.43 $\mu g/m^3$, 27.94±13.61 $\mu g/m^3$ and 30.38±11.38 $\mu g/m^3$ in marine, terrestrial and mixed air masses, respectively. Notably, $NO_3^-$ had the highest proportion (26.94%, 6.4%~52.6%), followed by $SO_4^{2-}$ (21.94%, 9.4%~33.4%) and $NH_4^+$ (20.26%, 5.8%~35.6%).*

22. L309: "with a range that from" should read "that ranged from" or "with a range of".

**Response of the authors:** We have revised the sentence in Line 318-319.

*A high concentration of Na⁺ was observed, with a range of 3.15±1.69 μg/m³, and accounted for 14.47% of the total water-soluble ions.*

23. L310: "ion" should read "ions".

**Response of the authors:** We have revised the sentence in Line 318-319.

*A high concentration of Na⁺ was observed, with a range of 3.15±1.69 μg/m³, which accounted for 14.47% of the total water-soluble ions.*

24. L315: "The concentration of K⁺ 0.24±0.20 μg/m³" should read "The concentration of K⁺ was 0.24±0.20 μg/m³".

**Response of the authors:** We have revised the sentence in Line 323-326.

*The concentration of K⁺ was 0.24±0.20 μg/m³ and 0.26±0.10 μg/m³ in the terrestrial and mixed air-mass samples, and was twice as high as those in the marine air-mass samples (0.11±0.05 μg/m³), which suggested an important contribution from anthropogenic emissions.*

25. L324: "coal combination in winter heating" probably did the authors mean "coal combustion"?

**Response of the authors:** We have revised the sentence in Line 330-332.

*Overall, from the composition and concentration of water-soluble ions in PM₂.₅, the coastal city was more affected by sea salt, coal combustion and dust events in early spring.*

26. L356: "in both terrestrial and mixed air-mass samples" should read "in both terrestrial and mixed air-masses samples".

**Response of the authors:** We have revised the sentence in Line 363-364.

*Cyanobacteria exhibit a higher concentration in both terrestrial and mixed air-masses samples.*

27. L366: please replace "including" by "included".

**Response of the authors:** We have revised the sentence in Line 374-376.

*These bacteria included a series of opportunistic pathogens and were found abundant in terrestrial and mixed air masses samples.*

28. L372: "to the previously studies" should read "to the previous studies".

**Response of the authors:** We have revised the sentence in Line 379-381.

*The dominant fungal phyla were Ascomycota (77.29%) and Basidiomycota (21.58%), which were similar to the previous studies (Du et al., 2018; Liu et al., 2019; Zeng et al., 2019).*

29. L388: I guess that "and automatic compound" should read "and aromatic compound".

**Response of the authors:** We have revised the sentence in Line 394-395.

*Marine air-mass samples were enriched with mammal gut bacteria, as well as hydrocarbon and aromatic compound degradation bacteria.*

30. L394: "the prevalence of Saprotroph fungi was observed higher in samples" should read "the observed prevalence of Saprotroph fungi was higher in samples".

**Response of the authors:** We have revised the sentence in Line 400-401.

*In particular, the observed prevalence of Saprotroph fungi was higher in samples from marine air masses, such as those containing Aspergillus.*

31. L406: there is one extra parenthesis.

**Response of the authors:** We have revised the sentence in Line 412-414.

*These include bacteria such as Bacillus, Streptococcus, and Deinococcus (Maki et al., 2010; Park et al., 2018; Qi et al., 2021).*

32. L408: "airmasses" should read "air masses".

**Response of the authors:** We have revised the sentence in Line 414-415.

*These microorganisms can adhere to particulate matter during long-distance transport of air masses after being released from their terrestrial habitats.*

33. L413: please replace "conducted" by "shown".

**Response of the authors:** We have revised the sentence in Line 420-421.

*Community disparities influenced by terrestrial, marine, and mixed air masses was shown in Table S2, S3, Fig. 3 and Fig. 4.*

34. L415: "air mass" should read "air mass samples".

**Response of the authors:** We have revised the sentence in Line 421-423.

*The Principal Coordinates Analysis (PCoA) revealed distinct clusters corresponding to terrestrial, marine, and mixed air mass samples (Fig. S4).*

35. L467: "automatic compound degradation bacteria" should read "aromatic compound degradation bacteria".

**Response of the authors:** We have revised the sentence in Line 472-474.

*Marine air-mass samples were enriched with mammal gut bacteria, hydrocarbon and aromatic compound degradation bacteria, and undefined Saprotroph fungi.*

36. L482: "diverse and abundance microbial populations" should read "diverse and abundant microbial populations".

**Response of the authors:** We have revised the sentence in Line 487-489.

*Air masses transported over long distances from the continent appear to harbor diverse and abundant microbial populations (Kakikawa et al., 2009; Deleon-Rodriguez et al., 2013).*

37. L488-490: "…accompanied the dust transportation to the downwind of Asian Dust including the coastal city of Weihai. Influenced by mixed air masses, bacterial community was 490 significantly positively correlated with $K^+$ (P<0.01) and $PM_{10}$ (P<0.05)". The sentence is weird.

**Response of the authors:** We have revised the sentence in Line 491-494.

*Dust-borne bacteria, such as Staphylococcus, Delftia, Pseudoalteromonas and Deinococcus, are likely introduced into the atmosphere during Asian Dust events. Most of these bacteria accompany the transportation of dust particles to the downwind coastal city.*

38. L491: "showed high positively correlated" should read "showed high positive correlation" or "were positively correlated".

**Response of the authors:** We have revised the sentence in Line 495-497.

*Similarly, microbial communities showed high positive correlation with ions from continental sources, such as $K^+$, $Mg^{2+}$, and $Ca^{2+}$ in terrestrial and mixed air mass samples.*

39. L511: Replace "Except" » by "Besides"?

**Response of the authors:** We have revised the sentence in Line 515-517.

*Besides anthropogenic sources, such as industrial emissions, the origins of $Mg^{2+}$ and $Na^+$ in coastal regions should also consider the impact of sea salt (Sun et al., 2022).*

40. L517: "temperature have" should read "temperature has".

**Response of the authors:** We have revised the sentence in Line 522-523.

*Influenced by mixed air masses, temperature has a greater impact on fungal community, which was positively correlated with Malasseziales and Davidiellaceae.*

41. L519: "marine airmass samples" should read "marine air mass samples".

**Response of the authors:** We have revised the sentence in Line 523-527.

*In marine air mass samples, a positive correlation between air temperature and certain microorganisms (Aerococcus, Cloacibacterium, Sphingobium, Enhydrobacterium, Davidiellaceae, Malasseziales) also indicated that the increase in air temperature in spring favors the survival of airborne microbes (Jones and Harrison, 2004).*